# StefaLand: An Efficient Geoscience Foundation Model That Improves Dynamic Land-Surface Predictions

## Abstract

Managing natural resources, meeting growing societal needs, and reducing risks from floods, droughts, wildfires, and landslides require models that can accurately predict climate-driven land–surface responses. Traditional models of environmental impact, whether process-based or task-specific machine learning, often struggle with spatial generalization because they are trained on limited observations and can degrade under concept drift. Recently proposed vision foundation models trained on satellite imagery demand massive compute, and they are often ill-suited for dynamic land surface prediction tasks. We introduce StefaLand, a generative spatiotemporal Earth foundation model centered on landscape interactions. Stefa-Land is demonstrated to improve predictions on four important tasks across five datasets: streamflow, soil moisture, soil composition and landslides, compared to previous state-of-the-art methods, showing especially strong ability to generalize across diverse landscapes, including data-scarce regions. The model builds on a masked autoencoder architecture that learns deep joint representations of landscape attributes, and its design reflects a deliberate integration of ideas adapted to geoscience. These include a location-aware architecture that fuses static and time-series inputs, an attribute-based rather than image-based representation that drastically reduces compute demands, and residual fine-tuning adapters that strengthen knowledge transfer across tasks. Their alignment with domain knowledge enables StefaLand to deliver robust performance on various dynamic land–surface tasks. StefaLand can be pretrained and finetuned on commonly-available academic compute resources compared with commercial foundation models, yet consistently outperforms state-of-the-art supervised learning baselines and fine-tuned vision foundation models. To our knowledge, this is the first geoscientific land-surface foundation model that demonstrably improves dynamic land surface interaction prediction tasks and supports a wide range of downstream applications.

## 1 Introduction

Climate change is ushering in strong and widespread changes on the land surface, including higher frequencies of floods, droughts, wildfires and other geohazards (Ebi et al., 2021; IPCC, 2021). To mitigate the impact of these disasters, there are urgent needs for models that can accurately predict land surface dynamics such as streamflow, soil moisture, soil composition, landslides, snow water equivalent, groundwater levels, and vegetation carbon content. Among these, soil moisture controls the partition of rainfall into infiltration and runoff, modulates flood generation and landslides, and critically influences land-atmosphere interactions (Dorigo et al., 2013a). Streamflow is the flow rate of water running in the rivers, the most accessible water resource to humans, and too high or too low streamflow can cause flooding or hydrologic drought, respectively. Soil composition (sand, silt, clay fractions) governs infiltration capacity and root-zone storage, while slope–soil-vegetation interactions directly influence landslide hazards. Here, we limit our scope to the predictions of dynamical or static land surface processes that represent the impacts of climate change. Predicting these dynamic variables is distinct from image-recognition tasks, as here we seek to predict what will happen in the near or distant future.

Traditionally, these tasks were undertaken by physics-based models that take atmospheric forcings (precipitation, temperature) as inputs and sequentially calculate the physical processes that eventually lead to the variables of interest (Li et al., 2015). In recent years, there has been a proliferation of data-driven machine learning (ML) models (Solomatine & Ostfeld, 2008). These models are often set up to accept forcing (dynamic weather) and landscape characteristics (static) data as inputs, and are trained to directly predict the natural land surface variables given the weather inputs. However, up to now, most of the geoscientific ML models have been supervised ML approaches trained specifically for a narrow set of tasks. A particularly notable gap is that currently established geoscience foundation models have largely been trained on satellite imagery for landcover-identification tasks (Jakubik et al., 2023; 2025; Schmude et al., 2024), which limits their ability to capture dynamic land–atmosphere interactions. As a result, valuable temporal datasets and ground-based observations remain underutilized (Xie et al., 2023), and to our knowledge no foundation model has yet been developed with a primary focus on dynamical land surface modeling.

A grand challenge for geoscientific ML models is to improve their spatial generalization, because a frequent issue facing them is the sparsity and spatial imbalance of observational data. Satellite data are often coarse in resolution and uncertain compared to in-situ measurements. SMAP, for example, provides global soil moisture observations at 9–36 km resolution every 2–3 days, which are useful for regional climate and hydrologic research but far less valuable than in-situ probes for operational field tasks such as irrigation scheduling or crop stress monitoring (Entekhabi et al., 2010). However, in-situ data, due to the cost of installing instruments and varying policies on data sharing, is only available in high density in certain regions of some developed nations. For example, streamflow gauge data are abundant in the United States, Europe, Australia and Japan, but remain sparse in Africa, South America, and much of Asia (Global Runoff Data Centre, 2020). Similar distribution patterns are found for high-quality in-situ soil moisture probes and soil property measurements. As quantified in many studies (Feng et al., 2023), a deep network trained on data from some regions can face substantial performance degradation when applied in data-scarce regions. This occurs partly because there are not enough sites to learn the true dependencies of the targets on static land surface characteristics, and partly because of systematic data discrepancies across regions (concept drift). While such limitations hinder traditional supervised ML models, foundation models offer a potential path forward: by jointly learning from broad, heterogeneous datasets (including temporal records and ground observations where available), they may transfer useful representations to data-scarce regions where task-specific training data are limited.

**Related Work:** In hydrologic and ecosystem predictions, supervised long short-term memory (LSTM) networks (Hochreiter & Schmidhuber, 1997) remain a highly popular architecture, in part because land surface processes often behave like Markov processes where LSTMs' gating mechanisms handle noisy continuous inputs well(Kratzert et al., 2018). Attempts to adopt transformers, so successful in natural language processing, have generally found it difficult to noticeably surpass LSTM in time series regression tasks (Xue et al., 2023; Liu et al., 2024), with evidence of overfitting on continuous signals (Zeng et al., 2022). Nonetheless, recent studies show that with task-specific modifications and careful fine-tuning, transformers can achieve competitive results in extreme event prediction (Wen et al., 2023), precisely the areas where current hydrologic models struggle most with spatial generalization.

Traditional hydrologic research on "prediction in ungauged basins" (PUB) have examined regionalization and spatial interpolation approaches including clustering or classifying catchments and transferring parameters from donor catchments in the same class (Hrachowitz et al., 2013; Yang et al., 2023). Such an expert-derived design represents a crude practice of unsupervised learning that indicates the importance of understanding the joint data distribution. However, modern weakly-supervised foundation models can, in general, much better grasp the joint data distribution than expert-driven approaches. Foundation models offer a promising approach to address these spatial generalization challenges. By pretraining on large-scale datasets to learn generalizable representations, these models can potentially transfer knowledge across regions and geoscientific domains (Zhang et al., 2024).

Existing geoscience foundation models, e.g., TerraMind (Jakubik et al., 2025) and Prithvi (Hsu et al., 2024) from IBM, and Aurora (Bodnar et al., 2025) from Microsoft, have largely focused on satellite imagery, which may not capture the temporal dynamics and physical processes most relevant to land surface predictions. For example, TerraMind is trained on 9 million globally distributed satellite imagery samples across various modalities (optical, radar, elevation, land use) to create a generative multimodal foundation model for Earth observation (Jakubik et al., 2023). It has been claimed to

serve as *"any-to-any generative, multimodal foundation model for Earth observation"* (Jakubik et al., 2025). However, many critical variables regarding soil and the subsurface are not directly observable from space. In addition, satellite images are noisy and a large portion of the data is redundant across repeated revisits or irrelevant for hydrologic processes. Therefore, it is unclear whether they are of high relevance to land-surface dynamical prediction tasks.

**Our contributions:** We present Spatial-Temporal Earth Foundation model with Attributes for the Land Surface (StefaLand), the first land-focused geoscientific foundation model designed for dynamic land–surface prediction. StefaLand improves predictions on streamflow, soil moisture, soil composition, and landslide susceptibility compared to state-of-the-art supervised learning baselines as well as finetuned satellite-image-trained earth foundation models. Especially, it shows strong spatial generalization across diverse landscapes and data-scarce regions for a wide variety of tasks. StefaLand's attribute-based rather than image-based design (with the potential to link to image-like inputs in the future) incorporates a variety of ground-based measurement data, emphasizes relevant land-surface physical processes, drastically reduces compute requirements while retaining global coverage, making it accessible to researchers with modest resources. Pretraining our model required only about 720 V100 GPU hours (could be shorter with more advanced GPUs). The model builds on a masked autoencoder backbone, a location-aware fusion of static and time-series inputs, grouped masking to promote cross-attribute interactions, and residual fine-tuning adapters, into a coherent design guided by geoscientific knowledge. Taken together, these contributions establish StefaLand as an efficient and accessible complement to vision-based foundation models.

## 2 METHODS

Dynamic land–surface prediction requires combining heterogeneous information: static landscape attributes such as topography, soils, vegetation, and geology, together with dynamic forcings such as precipitation and temperature. StefaLand addresses this challenge with a transformer-based masked autoencoder that jointly embeds static and dynamic variables, pretrained with a cross-variable masking strategy, and then adapted for specific prediction tasks with task-specific heads.

### 2.1 STEFALAND TRANSFORMER ARCHITECTURE

**Embedding.** Each dynamic input variable $c$ at time $t$ is independently embedded with a nonlinear two-layer projection:

$$z_{t,c} = \text{GELU}(x_{t,c}W_{1,c} + b_{1,c})W_{2,c} + b_{2,c}, \tag{1}$$

where $W_{1,c} \in \mathbb{R}^{1 \times 64}$ and $W_{2,c} \in \mathbb{R}^{64 \times 256}$. Summing across all $C$ dynamic variables yields a per-step embedding:

$$z_t = \sum_{c=1}^{C} z_{t,c}. \tag{2}$$

Static attributes $s_i$ are embedded similarly:

$$z_{\text{static},i} = \text{GELU}(s_i W_{1,i} + b_{1,i})W_{2,i} + b_{2,i}, \tag{3}$$

and concatenated as a special static token, producing the joint sequence:

$$Z = [z_1; z_2; \dots; z_T; z_{\text{static}}]. \tag{4}$$

**Cross-Variable Group Masking (CVGM).** Let $G = \{g_1, \dots, g_K\}$ be groups of related variables. During pretraining, a temporal window $[\tau, \tau + \ell)$ is sampled, and entire groups are stochastically masked:

$$m_k \sim \text{Bernoulli}(p_{\text{mask}}). \tag{5}$$

For all $c \in g_k$ with $m_k = 1$, the embedding is replaced by a learned mask vector $\mathbf{m}_c \in \mathbb{R}^{256}$. This forces reconstructions to rely on cross-variable dependencies rather than treating drivers independently.

**Transformer encoder.** After adding learnable positional encodings $P$, we obtain:

$$\tilde{Z} = Z + P. \tag{6}$$

This sequence is processed by $N$ Transformer blocks:

$$A^{(\ell)} = \text{MHA}(H^{(\ell-1)}), \tag{7}$$

$$\tilde{H}^{(\ell)} = \text{LayerNorm}(H^{(\ell-1)} + A^{(\ell)}), \tag{8}$$

$$F^{(\ell)} = \text{FFN}(\tilde{H}^{(\ell)}), \tag{9}$$

$$H^{(\ell)} = \text{LayerNorm}(\tilde{H}^{(\ell)} + F^{(\ell)}). \tag{10}$$

**Decoder and loss.** The final hidden states are passed through a bidirectional LSTM to refine local temporal continuity:

$$U = \text{LSTM}(H^{(N)}W_{\text{proj}} + b_{\text{proj}}). \tag{11}$$

Outputs corresponding to masked groups are projected back to the original variable dimensions. The reconstruction loss is

$$\mathcal{L} = \sum_{c \in \mathcal{M}} w_c \frac{\|\hat{x}_c - x_c\|_2^2}{\sigma_c^2}, \tag{12}$$

where $\mathcal{M}$ is the set of masked variables, $\sigma_c$ is the standard deviation of variable $c$, and $w_c$ is a learnable weight.

## 2.2 PRETRAINING DETAILS

The pretraining dataset is a derived global attribute dataset spanning $\sim$8,000 locations (basins) over 40 years. Variables were chosen to represent the key controls on fluxes of water, energy, momentum, sediment, and nutrients. A complete list of variables, their group assignments, and their sources is provided in Appendix C.

The CVGM is designed so variables with reciprocal or bidirectional causality are masked together, preventing them from acting as predictors for each other. Most groupings are straightforward, such as masking silt and clay percentages together. One exception is soil depth, which we classify as a terrain attribute since it is strongly tied to topographic derivations. By masking and reconstructing at the group level, the model is encouraged to capture cross-domain interactions, such as the coupling between soil texture and climate seasonality. While the grouped-masking strategy has sometimes been employed in multimodality AI models, we did not find it applied in earth foundation models like Terramind, Prithvi, or Aurora, which instead use random dropout, spatial patches, or future periods for masking. Our objective is a reconstruction loss on the masked slice of the sequence, normalized by variable-wise standard deviations when available and with per-variable weighting via a learnable weight vector.

## 2.3 FINETUNING FOR PREDICTION TASKS

Our primary finetuning model, **StefaLand-resConn**, integrates pretrained embeddings with raw forcings through a residual pathway (Figure D1). Let $E_t$ denote the transformer embedding at time $t$, and $x_t$ the raw forcings. We compute:

$$r_t = f_{\text{conv+linear}}(x_t), \tag{13}$$

$$h_t = \text{LSTM}(E_t + r_t), \tag{14}$$

$$\hat{y}_t = W_o h_t + b_o. \tag{15}$$

Skip connections propagate $E_t$ and $r_t$ into intermediate layers, allowing general knowledge (from $E_t$) to be iteratively refined with task-specific signals ($r_t$). This design strengthens spatial generalization while adapting to local variability. To see a visual representation of this D1.

## 2.4 FOUNDATION MODEL COMPARISONS

For completeness, we also evaluated two existing Earth-Observation-oriented foundation models, TerraMind and PrithviWxC (Jakubik et al., 2025; Hsu et al., 2024), using the same fine-tuning

heads and training protocols as StefaLand. We coupled them together with our residual connection architecture and finetuned the added units while keeping the foundation models' weights frozen. Because both TerraMind and PrithviWxC require orders of magnitude larger storage space, data transfer, and fine-tuning costs than StefaLand, we chose comparison tasks selectively, focusing on cases where their pretraining data was potentially most relevant. Accordingly, we compared both models on soil moisture, while streamflow was included only in a more exploratory capacity (only TerraMind). For a rough comparison, StefaLand, TerraMind and PrithviWxC used respectively $\approx 2$, $\approx 11$ and $\approx 27$ TB data during pretraining. Due to the intensive data requirements for PrithviWxC, only surface-level variables likely relevant to land surface interactions, along with all 14 static variables, were used. The full atmospheric variables at differing pressure levels were excluded.

## 3 EXPERIMENTS

We tested the value of foundation model pretraining on 5 datasets and 6 experiments, including, streamflow on the CAMELS dataset on USA, CAMELS streamflow prediction with hybrid model, global streamflow, global in-situ soil moisture, global soil properties, and landslide susceptibility in Oregon, USA. For all experiments, hyperparameters were tuned with Ray Tune and kept consistent across model configurations within each experimental case (e.g., CAMELS streamflow, soil moisture, etc). Because we compare spatial generalization, we used temporal validation splits for hyperparameter optimization. Complete details of hyperparameters, forcings, and static features for all four experiments are in Appendix C.

**Model variants used.** Unless noted otherwise, pretrained encoders are *frozen* and only task heads are trained. We evaluate: (i) LSTM-SL: supervised LSTM baseline; (ii) StefaLand-direct: the StefaLand encoder together with its original decoder trained directly on the task data, without pretraining; (iii) StefaLand-resConn: pretrained encoder with residual pathway + LSTM decoder, our proposed architecture; (iv) StefaLand-noResConn: pretrained encoder with a simple adapter (no residual/LSTM); (v) StefaLand-scratch: same resConn head but StefaLand initialized randomly (no pretraining) and unfrozen (ablation test); (vi) TerraMind-resConn and PrithviWxC-resConn: EO/atmospheric foundation encoders frozen with the same residual head.

### 3.1 CAMELS STREAMFLOW PREDICTION

To compare spatial generalization on a well benchmarked dataset, we follow (Feng et al., 2021), testing prediction in ungauged basins (PUB) and ungauged regions (PUR). These correspond to randomized spatial K-fold and regional-specific K-fold regimes, respectively. We use CAMELS (Addor et al., 2017; Newman et al., 2014), restricted to the 531-basin subset with clear watershed boundaries (Newman et al., 2017). Basins were divided into 10 random groups for PUB and 7 contiguous regions for PUR, employing leave-one-out in both cases. To avoid leakage, all CAMELS-overlapping stations were removed during pretraining for PUB, and entire regions were excluded for PUR.

Table 1: CAMELS Streamflow PUB and PUR Results

| Model | Random holdout (ungauged basins) | | | | Regional holdout (ungauged regions) | | | |
|---|---|---|---|---|---|---|---|---|
| | RMSE ↓ | μbRMSE ↓ | Corr ↑ | NSE ↑ | RMSE ↓ | μbRMSE ↓ | Corr ↑ | NSE ↑ |
| LSTM - SL | 1.402 | 1.360 | 0.762 | 0.636 | 1.609 | 1.457 | 0.743 | 0.554 |
| StefaLand - direct | 1.882 | 1.849 | 0.538 | 0.395 | 1.982 | 1.949 | 0.230 | 0.201 |
| StefaLand - resConn | **1.111** | **1.068** | **0.869** | **0.717** | **1.344** | **1.334** | **0.801** | **0.635** |
| StefaLand - no resConn | 1.171 | 1.154 | 0.823 | 0.706 | 1.376 | 1.356 | 0.798 | 0.610 |
| StefaLand - scratch | 1.355 | 1.332 | 0.801 | 0.661 | 1.516 | 1.378 | 0.771 | 0.560 |
| TerraMind - resConn | 1.332 | 1.301 | 0.777 | 0.637 | 1.420 | 1.398 | 0.763 | 0.551 |

The foundation model pretraining clearly provides a rare boost to generalization capability compared to supervised learning models (LSTM-SL and StefaLand-direct) across both PUB and PUR (Table 1). StefaLand-resConn's RMSE is almost 20% lower than that of supervised LSTM in PUB and 17% lower in PUR, while the correlation are noticeably higher. The value of pretraining is confirmed by

StefaLand-scratch, which appears to have only minor advantage compared to LSTM. The residual-connection architecture enables effective integration of transformer features with temporal dynamics, showing a modest benefit compared to StefaLand-no resConn, but both are clearly stronger than either LSTM or StefaLand-direct. Finally, TerraMind-resConn has quite comparable metrics to LSTM; the pretrained TerraMind offers no extra generalization value in this case.

We ran additional experiments that hybridize StefaLand with the HBV1.1 physics backbone on the same PUB/PUR splits, testing its ability to parameterize physics-based models. These hybrids achieved up to a 13% RMSE reduction and a 10% correlation gain compared to the LSTM–HBV1.1 baseline, demonstrating that pretraining strengthens hybridization with process models. By constraining predictions with physics while leveraging StefaLand features, these hybrids further improve upon the general results above and highlight the versatility of the approach. Full results are provided in Appendix B, Table B1.

On a related note, the supervised LSTM is not an easy benchmark to surpass. The original multi-basin LSTM and subsequent large-scale comparisons (Kratzert et al., 2019; Feng et al., 2021) showed that vanilla Transformers generally fail to outperform LSTMs on rainfall–runoff prediction (Liu et al., 2024, Table 1 therein). The LSTM NSE values reported here are very similar to those in the domain literature (Feng et al., 2021). Broader evaluations likewise continue to rank LSTM-family models among the best-performing approaches (Lees et al., 2021), and even Google's global flood-forecasting system adopts an encoder–decoder LSTM backbone (Nearing et al., 2024).

## 3.2 GLOBAL STREAMFLOW

We designed a global-scale runoff prediction experiment to assess robustness and generalization worldwide. We filtered global basin datasets based on data completeness and retained 3,434 basins. To manage computational cost, we employed random hold-out sampling with three-fold cross-validation. Additionally, we implemented a regionally hold-out continental scenario (RH-C), excluding all basins from North America, South America, and Europe, respectively, from training, then evaluating on the excluded continent.

Table 2: Global streamflow prediction across 3,434 basins worldwide

| Model | Random holdout (ungauged basins) | | | Regional holdout (ungauged continents) | | |
|---|---|---|---|---|---|---|
| | RMSE ↓ | μbRMSE ↓ | Corr ↑ | RMSE ↓ | μbRMSE ↓ | Corr ↑ |
| LSTM - SL | 0.870 | 0.864 | 0.798 | 1.253 | 1.202 | 0.672 |
| StefaLand - direct | **0.749** | **0.751** | **0.843** | **1.075** | **1.048** | **0.697** |
| TerraMind - resConn | 1.156 | 1.111 | 0.580 | 1.234 | 1.158 | 0.670 |

Cross-continental transfer testing using three-fold validation. Detailed metric calculations in Appendix E.

Results indicate that StefaLand-direct outperformed both LSTM-SL and the fine-tuned vision-based Transformer model (TerraMind) across all metrics for either random or regional holdout (Table 2). Specifically, in the random holdout scenario, StefaLand achieved an RMSE of 0.749, approximately 14% lower than LSTM (0.87), with Corr improving to 0.843. Notably, StefaLand's unbiased RMSE closely matched its RMSE, indicating errors were primarily random fluctuations rather than systematic bias. In contrast, TerraMind exhibited a higher RMSE of 1.156 and a low Corr of 0.580. We hypothesize this is because the pretraining satellite image data for the vision foundation network TerraMind did not have high relevance to hydrologic predictions like streamflow. We note that using TerraMind here was exploratory, as it was not specifically designed or pretrained for this use case.

Under the more challenging RH-C scenario, all models exhibited larger errors, but StefaLand continued to perform best. Its RMSE was 1.075, approximately 14.1% lower than LSTM's 1.253, while also maintaining the highest Corr (0.697). Moreover, StefaLand's ubRMSE remained close to its RMSE, confirming robust correction of regional-scale biases even under extreme out-of-domain conditions. Conversely, LSTM and TerraMind showed larger gaps between ubRMSE and RMSE, highlighting challenges in producing hydrologically-relevant features.

### 3.3 GLOBAL SOIL MOISTURE

We evaluated finetuning StefaLand for soil moisture predictions following Liu et al. (2023a), using ISMN (Dorigo et al., 2011; 2013a). Even though there is a globally covering satellite-based product for soil moisture, the data quality can hardly match that of in-situ moisture sensors; thus the ability to generalize in-situ data is valuable. ISMN consists of 1,316 ground-based stations. We performed five-fold spatial cross-validation for random holdout and a regional holdout on Europe, training on all other continents while excluding European sites (129) for testing. We tested StefaLand-direct, StefaLand with and without residual connections (resConn / noResConn), a from-scratch ablation, an LSTM baseline, and a version using IBM's TerraMind encoder with resConn. LSTM again serves as the established state-of-the-art baseline (Wang et al., 2024; Liu et al., 2023b).

Table 3: Soil moisture prediction across 1,316 ISMN stations

| Model | Random location holdout (random sites) | | | Regional holdout (Europe) | | |
|---|---|---|---|---|---|---|
| | RMSE ↓ | μbRMSE ↓ | Corr ↑ | RMSE ↓ | μbRMSE ↓ | Corr ↑ |
| LSTM - SL | 0.073 | 0.055 | 0.764 | 0.112 | **0.053** | 0.510 |
| StefaLand - direct | 0.140 | 0.103 | 0.637 | 0.135 | 0.112 | 0.503 |
| StefaLand - resConn | **0.068** | **0.054** | **0.783** | **0.090** | 0.059 | **0.638** |
| StefaLand - no resConn | 0.075 | 0.057 | 0.741 | 0.095 | 0.058 | 0.545 |
| StefaLand - scratch | 0.074 | 0.058 | 0.749 | 0.108 | 0.064 | 0.528 |
| TerraMind - resConn | 0.083 | 0.062 | 0.694 | 0.101 | 0.080 | 0.519 |
| PrithviWxC - resConn | 0.081 | 0.060 | 0.703 | 0.103 | 0.079 | 0.523 |

5-fold spatial validation and cross-continental validation on Europe (129 sites). Detailed metric calculations in Appendix E.

Against these strong baselines, the soil moisture experiments confirm the superiority of the StefaLand-resConn architecture (Table 3). It achieves the best performance, with RMSE of 0.068 and correlation of 0.783 for random holdout, and maintains strong performance even in cross-continental testing on Europe (RMSE = 0.090, Corr = 0.638). The direct StefaLand model performs poorly because it lacks an effective mechanism to capture task-specific temporal dependencies. Similar to the streamflow cases above, TerraMind and PrithviWxC showed no performance benefit compared to LSTM, underscoring the challenge of repurposing vision foundation models across domains and the importance of pretraining with geoscience-relevant variables. The regional holdout on Europe further demonstrates StefaLand-resConn's superior spatial generalization, achieving a 25% correlation improvement over the LSTM baseline in this difficult extrapolation scenario.

We want to emphasize that TerraMind and PrithviWxC were not designed for the hydrologic and land–surface prediction tasks studied here. They excel in their intended domains of Earth observation imagery and atmospheric variables, but are out-of-domain for dynamic land–surface and hydrologic modeling. We include them only to explore whether such models transfer any useful signal in our setting. These results should not be interpreted as evidence against their capability in other applications, but rather as a reflection of the mismatch between their pretraining objectives and the land–surface tasks we target. Nevertheless, we believe these benchmarks are helpful for clarifying their respective strengths, since the general AI community may not be familiar with these datasets and models that have been marketed as 'any-to-any' generative EO foundation models. Ultimately we view StefaLand as complementary to satellite-image foundation models, targeting a different but relevant data regime. We also could not identify other foundation models for land surface.

### 3.4 SOIL PROPERTY PREDICTION

There are different soil datasets, each collected with different protocols and data processing techniques, resulting in significant discrepancies. In this test, we finetuned StefaLand to predict in-situ soil profile data from another dataset (ISRIC). This application can produce a seamless dataset that is consistent with a set of in-situ data, improving data availability and addressing systematic biases. In addition, it helps us understand the noise associated with each dataset. StefaLand's pretraining soils dataset is HWSD, which has some overlap but also extensive differences from ISRIC, which is larger and

potentially noisier. We finetuned StefaLand to predict one soil texture property (e.g., clay percentage) in ISRIC while masking the corresponding complementary attribute (e.g., sand) from the same profile to avoid information leakage, probing how easy it is to infer soil properties using other attributes such as climate, terrain and land cover. We compared StefaLand with and without pretraining against a supervised random forest baseline. Train-test splits can be found here C.2

Table 4: In-situ soil property prediction using ISRIC WoSIS data.

| Model | WoSIS Property | Corr ↑ | $R^2$ ↑ |
|---|---|---|---|
| StefaLand - direct | Clay | 0.197 | 0.038 |
| StefaLand - finetune | Clay | **0.509** | **0.259** |
| Random Forest | Clay | 0.456 | 0.207 |
| Linear Regression | Clay | 0.138 | 0.019 |
| StefaLand - direct | Sand | 0.253 | 0.064 |
| StefaLand - finetune | Sand | **0.704** | **0.495** |
| Random Forest | Sand | 0.585 | 0.342 |
| Linear Regression | Sand | 0.347 | 0.120 |

StefaLand with finetuning achieved markedly higher predictive power for both clay and sand fractions, substantially outperforming direct training, random forest, and linear regression baselines. The other models struggle in this task, with random forest (RF) scoring noticeably lower. This task is not dynamical prediction, but a test of the models ability to build deep representations of the landscape that can be adapted to infer certain attributes using cross-variable group connections. StefaLand-finetune's advantage against random forest suggests there are indeed deep representations uncovered by pretraining but not exploited by random forest. Finetuning StefaLand is also apparently superior to training from scratch as supervised learning (StefaLand - direct), again highlighting the value of pretraining. These results highlight StefaLand's ability to reconcile disparate attribute datasets and improve the utility of noisy in-situ observations.

### 3.5 LANDSLIDE SUSCEPTIBILITY PREDICTION

Landslide is a geohazard that kill thousands each year. We next evaluated StefaLand for landslide susceptibility prediction using the SLIDO dataset from the State of Oregon, which provides detailed landslide occurrence records. Following Liu et al. (2025), this is a binary classification task indicating the presence or absence of landslides in a 30m by 30m patch. We finetuned StefaLand by extracting frozen hidden features and concatenating them with a 2D CNN, then retrained the CNN classifier to assess StefaLand's ability to provide generalizable geoscience features.

Table 5: Landslide susceptibility prediction results on the Oregon SLIDO dataset.

| Model | Accuracy ↑ | Precision ↑ | Recall ↑ | F1 ↑ | ROC AUC ↑ |
|---|---|---|---|---|---|
| Logistic Regression | 0.742 | 0.707 | 0.792 | 0.47 | 0.819 |
| Random Forest | 0.751 | 0.703 | **0.834** | 0.763 | 0.839 |
| CNN2D | 0.778 | 0.760 | 0.787 | 0.773 | 0.863 |
| StefaLand + CNN2D | **0.799** | **0.791** | 0.793 | **0.792** | **0.875** |

*Note*: All baseline results (Logistic Regression, Random Forest, CNN2D) are taken from previously published 30m-resolution experiments in Liu et al. (2025), except for StefaLand + CNN2D, which represents our proposed method.

Results show that StefaLand's pretrained features improved the CNN's generalization, yielding modest gains across all metrics except Recall. This is a particularly difficult baseline to improve so even modest gains are rare. ROC AUC increased from 0.863 to 0.875, and precision rose from 0.760 to 0.791, reflecting fewer false positives. While random forest finds a high Recall, this is a particular realization with a large tradeoff on Precision. StefaLand produces overall well-rounded predictions, with both better Precision and Recall than CNN2D.

## 4 DISCUSSION

### 4.1 KEY FINDINGS AND CONTRIBUTIONS

As we review the literature, the methods to improve spatial generalization to data-scarce regions are rare and rarely effective (Beery et al., 2018; Gacu et al., 2025). In contrast, StefaLand, combined with lightweight fine-tuning heads, achieves state-of-the-art or competitive performance across four broad problem classes: streamflow (both CAMELS and global), soil moisture, soil composition, and landslide susceptibility, while also strengthening the parameterization of differentiable process-based models. Across tasks, the strongest gains come from architectures that fuse StefaLand embeddings with explicit temporal modeling via residual connections, indicating that pretraining on attribute-based spatiotemporal structure yields problem-relevant representations while temporal heads resolve sequence dynamics. These outcomes support the premise that foundation models can democratize prediction quality in data-scarce regions by improving out-of-domain transfer.

The five dynamic prediction cases, along with benchmark models that reproduce state-of-the-art results in the literature, together paint a clear picture. The pretraining of StefaLand on attributes builds deep landscape representations. These features are highly relevant to hydrologic prediction tasks and can avoid overfitting compared to those built in task-specific supervised learning (LSTM or StefaLand-direct), improving model spatial generalization. They are also decidedly more relevant to such tasks than those obtained from existing Earth Foundation models expensively trained on massive amounts of satellite images. It shows that the importance of larger pretraining data may not necessarily exceed that of problem relevance, and image-like data may not be the optimal data representation for such tasks. We stress that our approach is complementary to satellite-based foundation models: where they exploit large-scale visual patterns, StefaLand focuses on problem-relevant attributes, offering an efficient and domain-specific alternative. We have tested a large number of adapter formulations. While we can certainly test more finetuning options, especially internal foundation model layers, the computational and data-storage demands already start to be limiting.

Our attribute-based approach is at least an order of magnitude more efficient than pixel-wise satellite transformers. Our underlying transformer has far fewer parameters (roughly 12 million) and avoids the heavy data management requirements of image-centric pretraining. TerraMind's larger configuration corresponds to about 7,680 GPU hours (Jakubik et al., 2025), PrithviWxC require roughly 23,040 GPU hours (Schmude et al., 2024) and Aurora required roughly 14,592 GPU hours (Bodnar et al., 2025). In contrast, StefaLand's attribute-based pretraining requires only about 720 GPU hours, making high-quality spatial generalization feasible on limited budgets. These figures do not include the large gaps in data-storage and transfer demands during pretraining (StefaLand, Terramind and PrithviWxC used $\approx 2$, $\approx 11$ and $\approx 27$ data as mentioned earlier). In fact, it is costly and resource-straining to run the satellite-imagery-focused foundation models, which impedes us from comparing with them in every case. Finally, to the best of our effort, we could not identify other earth foundation models that are designed for land-surface predictions.

### 4.2 LIMITATIONS AND FUTURE WORK

Several limitations remain. The selection of geological- or ecologically-focused attributes is limited and more can be added to further characterize the subsurface. Two-dimensional (or image-like) data like elevation map can be selectively incorporated using vision transformer heads in the future. Expanding the range of targets to include variables such as evapotranspiration, snow water equivalent, and groundwater levels would broaden its applicability. Methodologically, advances such as uncertainty-aware prediction heads, and tighter integration with additional process models offer promising avenues to improve calibration and interpretability while preserving efficiency. Overall, StefaLand shows that attribute-centric pretraining combined with lightweight temporal or physics heads can deliver strong spatial generalization across geoscientific tasks while remaining computationally accessible. This points toward a practical path for high-quality predictions in regions where they are most needed but data are most limited.

## 5 REPRODUCIBILITY STATEMENT

The pretrained StefaLand model and all code for both pretraining and finetuning is released publicly at [https://anonymous.4open.science/r/StefaLand-9421/]. All datasets used in this work both pretraining dataset and all finetuneing benchmarks are fully public. A complete list of variables used for each task, along with their data sources, is provided in Appendix C. We also report all hyperparameters and model details in the same Appendix.

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

## A   DETAILED MODEL ARCHITECTURE

This appendix provides the complete mathematical formulation of the StefaLand model architecture more detailed then 2.

### A.1   EMBEDDING DYNAMIC AND STATIC INPUTS

StefaLand independently embeds each dynamic and static variable into a latent space. Specifically, for each dynamic variable $c$ at each time step $t$, a two-step nonlinear embedding is applied individually:

$$z_{t,c} = \text{GELU}(x_{t,c}W_{1,c} + b_{1,c})W_{2,c} + b_{2,c} \tag{16}$$

where $W_{1,c} \in \mathbb{R}^{1\times 64}$ and $W_{2,c} \in \mathbb{R}^{64\times 256}$ are embedding parameters. After embedding all dynamic variables individually, embeddings are stacked and summed across the variable dimension, resulting in a single embedding vector per time step:

$$z_t = \sum_{c=1}^{C} z_{t,c} \tag{17}$$

Similarly, static attributes are embedded individually:

$$z_{\text{static},i} = \text{GELU}(s_i W_{1,i} + b_{1,i})W_{2,i} + b_{2,i} \tag{18}$$

where separate embedding layers are used for static features. These individual static embeddings are then concatenated with dynamic embeddings along the temporal dimension, resulting in a unified embedding tensor:

$$Z = [z_1; z_2; \ldots; z_T; z_{\text{static}}] \tag{19}$$

This static embedding acts as a global learnable token, allowing the model to incorporate basin-specific context into temporal dynamics at any depth of the Transformer layers.

### A.2   LOCATION-AWARE CROSS-VARIABLE GROUP MASKING

StefaLand introduces Cross-Variable Group Masking (CVGM), a masking strategy that forces the model to capture interactions among correlated hydrologic variables rather than treating them independently. Given a predefinition of hydrological variables into groups $G = \{g_1, g_2, \ldots, g_k\}$, masking occurs as follows:

1. A temporal masking window $[\tau, \tau + \ell)$ is randomly sampled with length $\ell \sim \mathcal{U}(L_{\min}, L_{\max})$.

2. For each variable group $g_k$, a Bernoulli mask indicator $m_k \sim \text{Bernoulli}(p_{\text{mask}})$ determines whether the group is masked.

3. For each time step $t$ within the masked temporal window and each variable $c$ belonging to masked groups, the embedded feature vector is replaced by a learned mask vector $\mathbf{m}_c$.

Each hydrologic variable $c$ has its own trainable mask embedding vector $\mathbf{m}_c \in \mathbb{R}^{256}$. This CVGM procedure creates reconstruction targets that require modeling cross-variable dependencies and physical interactions.

### A.3   LEARNABLE POSITIONAL ENCODING

To provide positional information, StefaLand employs learnable positional encoding. Each position $i$, corresponding to each time step and the appended static embedding, is assigned a trainable embedding vector $\mathbf{p}_i$. The encoded embedding becomes:

$$\tilde{Z} = Z + P \tag{20}$$

where $P = [\mathbf{p}_1; \ldots; \mathbf{p}_{T+1}]$.

## A.4 TRANSFORMER ENCODER

The embeddings enriched by positional encoding are processed through an $N$-layer Transformer encoder, where each Transformer block successively applies Multi-Head Self-Attention (MHA) with $h$ attention heads, followed by a residual connection and Layer Normalization. Subsequently, a position-wise Feedforward Network (FFN) is applied, also followed by another residual connection and Layer Normalization:

$$A^{(\ell)} = \text{MHA}(H^{(\ell-1)}) \tag{21}$$

$$\tilde{H}^{(\ell)} = \text{LayerNorm}(H^{(\ell-1)} + A^{(\ell)}) \tag{22}$$

$$F^{(\ell)} = \text{FFN}(\tilde{H}^{(\ell)}) \tag{23}$$

$$H^{(\ell)} = \text{LayerNorm}(\tilde{H}^{(\ell)} + F^{(\ell)}) \tag{24}$$

## A.5 RECONSTRUCTION OF ORIGINAL INPUTS

The final hidden states from the Transformer encoder, $H^{(N)}$, are linearly projected and passed through a single-layer bidirectional LSTM to capture the local temporal dependencies and continuity:

$$U = \text{LSTM}(H^{(N)} W_{\text{enc-proj}} + b_{\text{enc-proj}}) \tag{25}$$

The outputs $U$ are then separated into dynamic and static components, $U_t$ and $U_{\text{static}}$, corresponding to the temporal sequence and static attributes:

$$U_t, U_{\text{static}} = U_{1:T}, U_{T+1} \tag{26}$$

Finally, both dynamic and static representations are individually projected back to their original dimensions through separate embedding layers, reconstructing the masked portions of the inputs. Dynamic variables are restored via:

$$\hat{x}_t = \text{DynamicDecEmbedding}(U_t) \tag{27}$$

while static attributes are restored by:

$$\hat{s} = \text{StaticDecEmbedding}(U_{\text{static}}) \tag{28}$$

The projections leverage the learned latent representations to reconstruct the original hydrologic inputs.

# B ADDITIONAL EXPERIMENTS

## B.1 PHYSICS-BASED DIFFERENTIABLE MODELING

To leverage domain knowledge and physical constraints inherent in hydrological systems, we implemented physics-based models that explicitly represent hydrological processes through mathematical formulations. These differentiable versions can be trained end-to-end within neural network frameworks, combining process understanding with machine learning flexibility (Shen et al., 2023).

For the process-based backbone, we employed the Hydrologiska Byråns Vattenbalansavdelning (HBV) model (Aghakouchak & Habib, 2010; Beck et al., 2020; Bergström, 1976; 1992; Seibert & Vis, 2012), a relatively simple bucket-type conceptual hydrologic model. HBV has state variables like snow storage, soil water, and subsurface storage, and can simulate flux variables such as evapotranspiration (ET), recharge, surface runoff, shallow subsurface flow, and groundwater flow. We used an updated modern version, HBV1.1 (Song et al., 2025), which includes modifications such as increased parallel storage components to represent heterogeneity within basins and dynamic parameterization capabilities.

The hybrid model employs a differentiable parameter learning (dPL) framework where neural networks generate parameters for HBV1.1, and errors are backpropagated through the entire system. A machine learning network takes basin attributes and meteorological forcings as inputs and outputs HBV parameters—both static (e.g., recession coefficients) and dynamic parameters that vary daily. Because HBV1.1 supports automatic differentiation, it serves as the physical backbone: during training, loss is calculated between simulated and observed streamflow, gradients are backpropagated through HBV equations, and neural network weights are updated. This differs from traditional calibration because parameters are learned regionally across all basins simultaneously rather than individually, allowing the network to capture generalizable relationships between basin characteristics and optimal parameters while maintaining mass balance constraints. The system uses 16 parallel response units for spatial heterogeneity and outputs diagnostic variables (e.g., evapotranspiration, soil moisture, baseflow) not directly trained on, providing interpretability with competitive performance.

For physics-based configurations, we tested: (1) a baseline LSTM–HBV1.1 configuration as a standard reference, (2) StefaLand HBV1.1 with resConn, which combines the physics-based approach with our residual connection architecture, and (3) StefaLand HBV1.1 without resConn. These physics-based approaches incorporate hydrological process understanding while maintaining the ability to learn from data..

Table B1: CAMELS Streamflow PUB and PUR Results (Physics-Based Models)

| Model | Random holdout (ungauged basins) | | | | Regional holdout (ungauged regions) | | | |
|---|---|---|---|---|---|---|---|---|
| | RMSE ↓ | µbRMSE ↓ | Corr ↑ | NSE ↑ | RMSE ↓ | µbRMSE ↓ | Corr ↑ | NSE ↑ |
| LSTM - HBV1.1 | 1.325 | 1.298 | 0.857 | 0.672 | 1.561 | 1.521 | 0.746 | 0.578 |
| StefaLand - resConn HBV1.1 | **1.234** | **1.216** | **0.863** | **0.714** | **1.345** | **1.332** | **0.842** | **0.643** |
| StefaLand - no resConn HBV1.1 | 1.315 | 1.302 | 0.848 | 0.707 | 1.401 | 1.379 | 0.835 | 0.623 |
| StefaLand Ablation - resConn HBV1.1 | 1.310 | 1.306 | 0.842 | 0.693 | 1.465 | 1.432 | 0.607 | 0.512 |

## B.2 LINEAR REGRESSION BASELINES

To justify the use of complex neural networks over traditional methods, we have conducted baseline comparisons using linear regression models. As shown in the table below, linear regression performs poorly across all tasks by a fair margin when compared to our neural network approaches.

Table B2: Additional experiments with linear regression baselines.

| Experiment | Random holdout | | | Regional holdout | | |
|---|---|---|---|---|---|---|
| | RMSE ↓ | µbRMSE ↓ | Corr ↑ | RMSE ↓ | µbRMSE ↓ | Corr ↑ |
| Camels Streamflow Linear Regression | 2.190 | 2.180 | 0.500 | 2.260 | 2.250 | 0.500 |
| Global Streamflow Linear Regression | 1.823 | 1.746 | 0.252 | 1.816 | 1.721 | 0.248 |
| Soil Moisture Linear Regression | 0.120 | 0.101 | 0.188 | 0.121 | 0.103 | 0.187 |

# C  EXPERIMENTAL DETAILS

Table C1: StefaLand Pretraining Configuration

| Parameter | Value |
|---|---|
| **General Settings** | |
| Task | pretrain |
| Model | MFFormer_dec_LSTM |
| Random seed | 111 |
| Time Period | 1980/1/1–2018/12/31 |
| **Sequence Configuration** | |
| Sequence length | 365 |
| Label length | 365 |
| Prediction length | 365 |
| Sampling stride | 1 |
| Minimum window size | 30 |
| Maximum window size | 90 |
| **Model Architecture** | |
| Input dimension (enc_in) | 32 |
| Decoder input (dec_in) | 6 |
| Output dimension (c_out) | 6 |
| Model dimension | 256 |
| Number of heads | 4 |
| Encoder layers | 4 |
| Decoder layers | 2 |
| Feed-forward dimension | 512 |
| Dropout | 0.1 |
| Activation | gelu |
| **Training Configuration** | |
| Optimizer | AdamW |
| Loss criterion | MaskedNSE |
| Epochs | 25 |
| Batch size | 256 |
| Learning rate | 0.0001 |
| Weight decay | 0.0 |
| Patience | 30 |
| Gradient clipping | 5.0 |
| Number of workers | 10 |
| **Loss Weights** | |
| Time series loss ratio | 1.0 |
| Static loss ratio | 0.5 |

Table C2: StefaLand Pretraining Variables and Sources

| Variable Type | Variable Name | Source |
|---|---|---|
| **Time Series Forcings** | Precipitation, Short-wave solar radiation downwards, Relative humidity, Maximum temperature, Minimum temperature, Potential evapotranspiration | Multi-Source Weather (MSWX) and Multi-Source Weighted-Ensemble Precipitation (MSWEP) (Beck et al., 2022; 2019) |
| **Static Attributes** | Forest cover fraction, grassland cover fraction | Climate Change Initiative (CCI) land cover dataset (ESA, 2017) |
| | Normalized Difference Vegetation Index (NDVI) | Terra Moderate Resolution Imaging Spectroradiometer (MODIS) Vegetation Indices (MOD13A3) (Didan, 2015a) |
| | Sand, silt, clay fractions | Harmonized World Soil Database (HWSD) (FAO et al., 2012) |
| | Elevation, slope, aspect | Global Multi-resolution Terrain Elevation Data (GMTED) (Danielson & Gesch, 2011; Ramcharan et al., 2018) |
| | Soil depth | Global 1-km Gridded Thickness of Soil, Regolith, and Sedimentary Deposit Layers (Pelletier et al., 2016) |
| | Carbonate sedimentary rock fraction | Global Lithological Map (GLiM) (Hartmann & Moosdorf, 2012) |
| | Rock porosity, permeability | GLobal HYdrogeology MaPS (GLHYMPS) (Gleeson et al., 2014) |
| | Population density | Gridded Population of the World (GPW) v4 dataset (CIESIN, 2016) |
| | GDP per capita; population density | Gross Domestic Product and Human Development Index over 1990-2015 (Kummu et al., 2018) |
| | Forest intact fraction | Intact Forest Landscapes Data (Potapov et al., 2017) |
| **Outputs** | None (self-supervised pretraining) | — |

Table C3: Attribute Groups Used in Group Masking Pretraining

| Group | Variables |
|---|---|
| **Topography** | meanelevation, meanslope |
| **Soil** | HWSD_clay, HWSD_sand, HWSD_silt, HWSD_gravel, SoilGrids1km_sand, SoilGrids1km_clay, SoilGrids1km_silt |
| **Geology** | permeability, Porosity, glaciers, permafrost |
| **Vegetation** | NDVI, FW |
| **Climate** | aridity, meanP, ETPOT_Hargr, meanTa, seasonality_P, seasonality_PET, snow_fraction, snowfall_fraction |

Table C4: CAMELS Streamflow HBV Model Hyperparameters

| Parameter | Value |
|---|---|
| **General Settings** | |
| Random seed | 111111 |
| Data sampler | finetune_sampler |
| **Training Configuration** | |
| Time period | 1989/10/01–2008/09/30 |
| Optimizer | Adadelta |
| Batch size | 64 |
| Epochs | 25 |
| **Neural Model Configuration** | |
| Sequence length | 365 |
| Hidden size | 512 |
| Dropout | 0.2 |
| Encoder layers | 4 |
| Decoder layers | 2 |
| Feed-forward dimension | 512 |
| **Physical Model (HBV-1.1)** | |
| Model type | HBV_1_1p |
| Number of runs (nmul) | 16 |
| Warm-up period | 365 days |
| Warm-up states | True |
| Dynamic dropout | 0.0 |
| Use routing | True |
| Dynamic parameters | parBETA, parK0, parBETAET |
| Near-zero threshold | 1e-05 |
| **Loss Function** | |
| Type | RmseLoss |

Table C5: CAMELS Streamflow Variables and Sources

| Variable Type | Variable Name | Source |
|---|---|---|
| **Time Series Forcings** | Precipitation, Temperature, Potential evapotranspiration, Solar radiation, Vapor pressure | Catchment Attributes and Meteorology for Large-sample Studies (CAMELS) (Addor et al., 2017; Newman et al., 2014) |
| **Static Attributes** | Elevation, slope, catchment area, forest cover, LAI, GVF, soil depth, porosity, conductivity, sand, silt, clay fractions, carbonate fraction, permeability, aridity, snow fraction, precipitation extremes | CAMELS |
| **Outputs** | Streamflow | CAMELS gauge records |

Table C6: Soil Moisture Model Configuration

| Parameter | Value |
|---|---|
| **General Settings** | |
| Mode | train_test |
| Random seed | 111111 |
| Data loader | onlylstm_loader |
| Data sampler | finetuning_noHBV |
| **Training Configuration** | |
| Time period | 2015/04/01–2020/12/31 |
| Target | soil_moisture |
| Optimizer | Adadelta |
| Batch size | 128 |
| Epochs | 50 |
| Save frequency | Every 25 epochs |
| **Neural Network Configuration** | |
| Hidden size | 128 |
| Dropout | 0.3 |
| Learning rate | 1.2 |
| Encoder layers | 16 |
| Decoder layers | 12 |
| Feed-forward dimension | 512 |
| Rho | 365 |
| **Loss Function** | |
| Type | RmseLoss |

Table C7: Soil Moisture Variables and Sources

| Variable Type | Variable Name | Source |
|---|---|---|
| **Time Series Forcings** | Albedo (BSA, WSA) | Moderate Resolution Imaging Spectroradiometer (MODIS) MCD43A3 version 6 (Schaaf, Crystal & Wang, Zhuosen, 2021) |
| | LST (Day, Night) | MODIS Land Surface Temperature/Emissivity Daily (MYD11A1) Version 6.1 (Wan et al., 2021) |
| | Precipitation | Global Precipitation Measurement (GPM), & MSWEP &ERA5 precipitation (Huffman et al., 2019; Beck et al., 2019; Muñoz Sabater, 2019) |
| | Forecast albedo, LAI (high/low vegetation), soil temperature (layer 1), surface pressure, solar radiation, 2 m temperature, evaporation, precipitation, U/V wind (10 m) | ECMWF Reanalysis v5 (ERA5) (Muñoz Sabater, 2019) |
| **Static Attributes** | elevation, slope, aspect, roughness, curvature | Global 1/5/10/100-km topography derivatives (Amatulli et al., 2018) |
| | Sand, clay, silt, bulk density | HWSD v1.2 (FAO et al., 2012) |
| | Land cover; urban; open water; snow/ice | ESA CCI Land Cover (ESA, 2017) |
| | NDVI | Vegetation Indices Monthly L3 Global 0.05Deg CMG (Didan et al., 2015) |
| **Outputs** | Soil moisture | International Soil Moisture Network (ISMN) (Dorigo et al., 2013b; 2011) |

Table C8: Global Streamflow Variables and Sources

| Variable Type | Variable Name | Source |
|---|---|---|
| **Time Series Forcings** | Precipitation, Temperature, Potential evapotranspiration, Radiation, Humidity | MSWX and MSWEP (Beck et al., 2022; 2019) |
| **Static Attributes** | Forest cover fraction, grassland cover fraction | Climate Change Initiative (CCI) land cover dataset (ESA, 2017) |
| | Normalized Difference Vegetation Index (NDVI) | Terra Moderate Resolution Imaging Spectroradiometer (MODIS) Vegetation Indices (MOD13A3) (Didan, 2015a) |
| | Sand, silt, clay fractions | Harmonized World Soil Database (HWSD) (FAO et al., 2012) |
| | Elevation, slope, aspect | Global Multi-resolution Terrain Elevation Data (GMTED) (Danielson & Gesch, 2011; Ramcharan et al., 2018) |
| | Soil depth | Global 1-km Gridded Thickness of Soil, Regolith, and Sedimentary Deposit Layers (Pelletier et al., 2016) |
| | Carbonate sedimentary rock fraction | Global Lithological Map (GLiM) (Hartmann & Moosdorf, 2012) |
| | Rock porosity, permeability | GLobal HYdrogeology MaPS (GLHYMPS) (Gleeson et al., 2014) |
| | Population density | Gridded Population of the World (GPW) v4 dataset (CIESIN, 2016) |
| | GDP per capita; population density | Gross Domestic Product and Human Development Index over 1990-2015 (Kummu et al., 2018) |
| | Forest intact fraction | Intact Forest Landscapes Data (Potapov et al., 2017) |
| **Outputs** | Streamflow | Global runoff data (GRDC, 2024) |

Table C9: Landslide (SLIDO, Oregon) Variables and Sources

| Variable Type | Variable Name | Source |
|---|---|---|
| **Input data** | Elevation | National Elevation Dataset (NED) (Gesch et al., 2018) |
| | Soil sand, silt, clay, bulk density, saturated hydraulic conductivity | Probabilistic Remapping of SSURGO (POLARIS) (Chaney et al., 2019) |
| | Lithology | Global Lithological Map (GLiM) (**?**) |
| | Rainfall | PRISM (PRISM Climate Group, 2014) |
| | NDVI | Moderate Resolution Imaging Spectroradiometer (MODIS) Vegetation Indices Monthly L3 (Didan, 2015b) |
| | Landcover | National Land Cover Database (NLCD) 2016 (Dewitz, 2019) |
| | Soil moisture | SMAP-HydroBlocks (SMAP-HB) (Vergopolan et al., 2021) |
| | slope, aspect, curvature, TWI, SPI | DEM-derived |
| **Outputs** | Landslide occurrence (binary) | Statewide Landslide Information Database for Oregon (SLIDO) (Franczyk, J. J et al., 2020) |

Table C10: Soil Composition (ISRIC) Variables and Sources

| Variable Type | Variable Name | Source |
|---|---|---|
| **Time Series Forcings** | Same as Table 7 | — |
| **Static Attributes** | Same as Table 7 | — |
| **Outputs** | Soil property (clay; sand; silt) | World Soil Information Service (WoSIS) (Batjes et al., 2020) |

Table C11: PrithviWxC Surface Variables and Sources (MERRA-2)

| Variable Type | Variable Name | Source |
|---|---|---|
| **Time Series Forcings** | Precipitation, fluxes, winds, soil moisture/temperature, LAI, runoff, etc. | NASA MERRA-2 reanalysis (Gelaro et al., 2017) |
| **Static Attributes** | Land/ocean/ice fractions, surface geopotential, subgrid orography | NASA MERRA-2 constants (Gelaro et al., 2017) |

Table C12: Computation Resources for StefaLand and Comparison Models

| Model | Seconds/Epoch | #GPUs | GPU Type | Memory |
|---|---|---|---|---|
| StefaLand (Pretraining) | 16,000 | 6 | NVIDIA V100 | 240 GB |
| StefaLand with resConn | 30 | 2 | NVIDIA V100 | 80 GB |
| StefaLand without resConn | 26 | 2 | NVIDIA V100 | 80 GB |
| LSTM Baseline | 12 | 2 | NVIDIA V100 | 80 GB |
| LSTM-HBV1.1 | 280 | 2 | NVIDIA V100 | 80 GB |
| StefaLand-resConn HBV1.1 | 320 | 2 | NVIDIA V100 | 80 GB |
| StefaLand-no resConn HBV1.1 | 300 | 2 | NVIDIA V100 | 80 GB |

Note: All values except pretraining are for the CAMELS benchmark experiment. The relative differences in computational requirements are consistent across other experiments.

## C.1 PRETRAINING DATA HANDLING

To handle large volumes efficiently, data were stored in shards rather than fully loaded into memory. At the start of each epoch, shards were randomly reindexed, providing diverse samples while avoiding costly materialization of the full dataset. This strategy ensures both reproducibility and wide sample coverage. The accompanying code release in the reproducibility statement enables reconstruction of this dataset and reproduces the full pretraining pipeline.

## C.2 DATASET SPLITTING

For the WoSIS soil dataset, we collected soil property data from 106,503 locations. After removing low-quality records (e.g., sand values greater than 1 or negative values), we randomly sampled 5,000 soil points to reduce computational cost. We then applied 5-fold cross-validation (k=5) on this subset.

For the landslide dataset at 30 m resolution, we used 14,604 historical landslide points. We split the dataset into 70% for training, 20% for validation, and 10% for testing.

# D  MODEL ARCHITECTURES

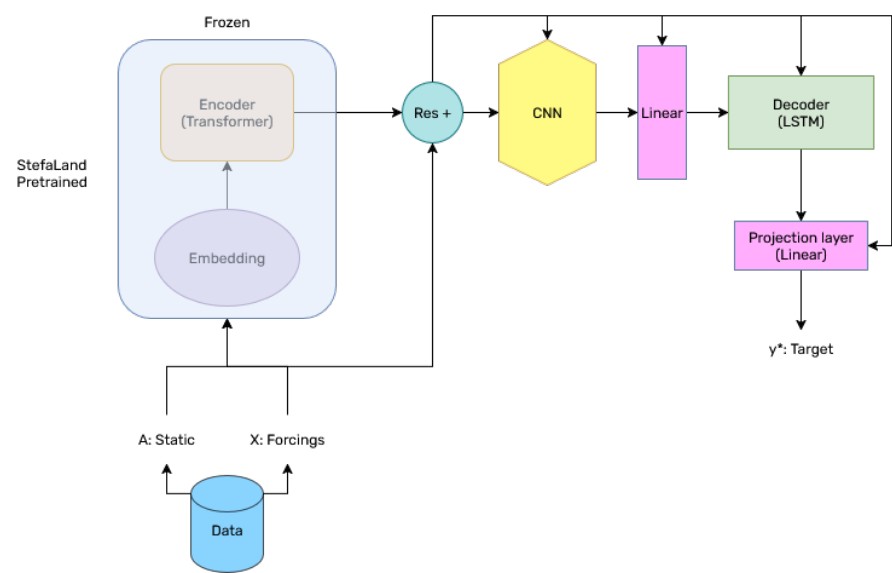

Figure D1: StefaLand model architecture with residual connections. The pretrained StefaLand encoder (frozen during fine-tuning) processes static attributes and generates embeddings that are combined with meteorological forcings through residual connections. This architecture enables iterative integration of transformer features with input data through the CNN module, allowing the model to effectively leverage pretrained representations while adapting to task-specific requirements.

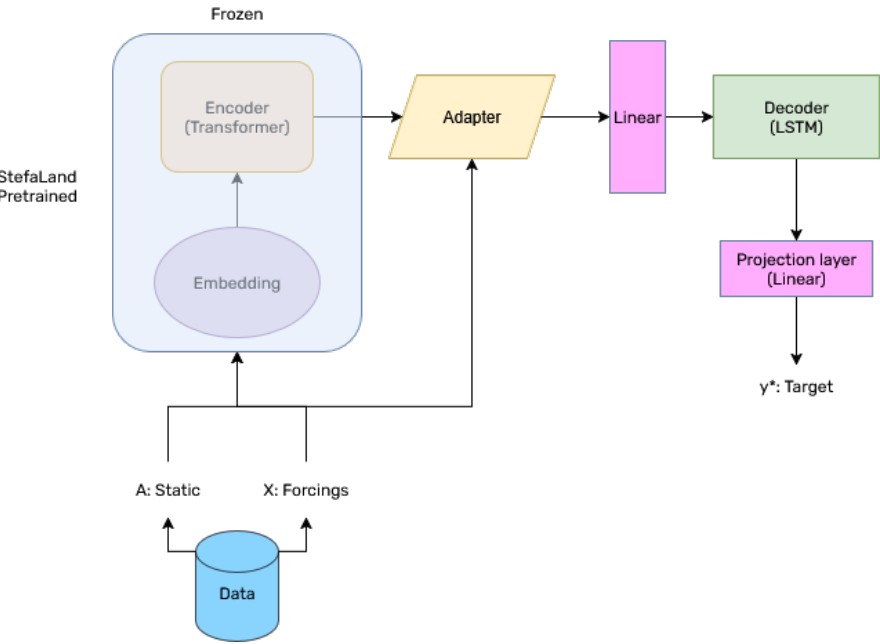

Figure D2: StefaLand model architecture without residual connections. In this configuration, the frozen transformer embeddings are used only once through a standard adapter that combines them with forcings and static features. This represents a more conventional fine-tuning approach where transformer features are not integrated iteratively throughout the decoding process.

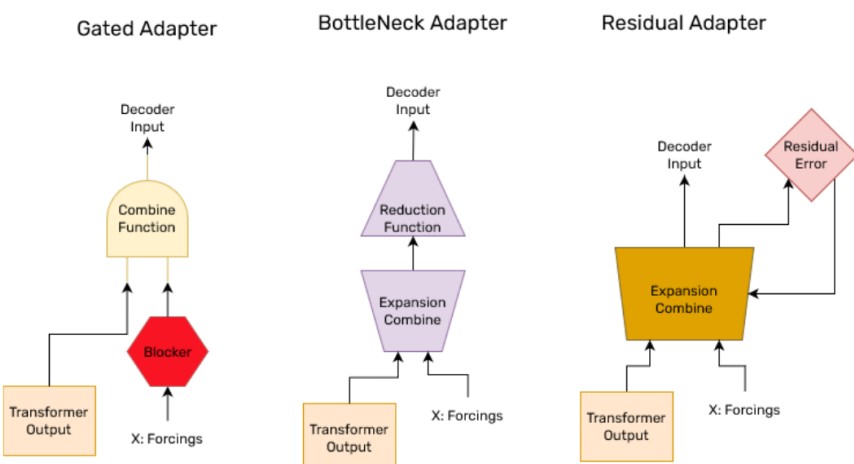

Figure D3: Different adapter architectures tested in our experiments. Left: Gated Adapter with a selector mechanism that controls information flow. Center: BottleNeck Adapter with compression and expansion phases. Right: Residual Adapter that adds transformer features through a skip connection.

# E  METRIC CALCULATIONS

This appendix details the calculation of the evaluation metrics used in our experiments. All metrics presented in the main paper tables are the median values across test basins or stations, as computed using the following formulations.

## E.1  PRIMARY EVALUATION METRICS

### E.1.1  ROOT MEAN SQUARE ERROR (RMSE)

RMSE measures the average magnitude of prediction errors. Lower values indicate better performance.

$$\text{RMSE} = \sqrt{\frac{1}{n} \sum_{i=1}^{n} (y_{\text{pred},i} - y_{\text{target},i})^2} \tag{29}$$

### E.1.2  UNBIASED ROOT MEAN SQUARE ERROR (µbRMSE)

µbRMSE removes the bias component from the error calculation, focusing on the error's random component. It is calculated by first computing anomalies from the mean for both predictions and targets.

$$y'_{\text{pred},i} = y_{\text{pred},i} - \overline{y}_{\text{pred}} \tag{30}$$

$$y'_{\text{target},i} = y_{\text{target},i} - \overline{y}_{\text{target}} \tag{31}$$

$$\text{µbRMSE} = \sqrt{\frac{1}{n} \sum_{i=1}^{n} (y'_{\text{pred},i} - y'_{\text{target},i})^2} \tag{32}$$

### E.1.3  CORRELATION (CORR)

Correlation quantifies the linear relationship between predictions and targets. Values range from -1 to 1, with 1 indicating perfect positive correlation.

$$\text{Corr} = \frac{\sum_{i=1}^{n} (y_{\text{pred},i} - \overline{y}_{\text{pred}})(y_{\text{target},i} - \overline{y}_{\text{target}})}{\sqrt{\sum_{i=1}^{n} (y_{\text{pred},i} - \overline{y}_{\text{pred}})^2 \sum_{i=1}^{n} (y_{\text{target},i} - \overline{y}_{\text{target}})^2}} \tag{33}$$

This is calculated using Pearson's correlation coefficient between predicted and observed values.

## E.2  SECONDARY METRICS

The following metrics are used in our comprehensive evaluation but may not appear directly in the main tables.

### E.2.1  NASH-SUTCLIFFE EFFICIENCY (NSE) / $R^2$

NSE evaluates the predictive skill relative to using the mean of observations as a predictor. Values range from $-\infty$ to 1, with 1 indicating perfect prediction.

$$\text{NSE} = 1 - \frac{\sum_{i=1}^{n} (y_{\text{target},i} - y_{\text{pred},i})^2}{\sum_{i=1}^{n} (y_{\text{target},i} - \overline{y}_{\text{target}})^2} \tag{34}$$

### E.2.2 MEAN ABSOLUTE ERROR (MAE)

MAE measures the average absolute difference between predictions and targets.

$$\text{MAE} = \frac{1}{n} \sum_{i=1}^{n} |y_{\text{pred},i} - y_{\text{target},i}| \tag{35}$$

### E.2.3 FLOW DURATION CURVE RMSE (RMSE_FDC)

RMSE_FDC evaluates errors in the statistical distribution of flows rather than in their timing.

$$\text{RMSE\_FDC} = \sqrt{\frac{1}{100} \sum_{j=1}^{100} (FDC_{\text{pred},j} - FDC_{\text{target},j})^2} \tag{36}$$

where $FDC_j$ represents the $j$-th percentile of the sorted flow values.

### E.2.4 FLOW BIASES

Several flow-specific biases were computed to evaluate performance across different flow regimes:

- $FLV$ (Low Flow Volume Bias): Percent bias in the lowest 30% of flows
- $FHV$ (High Flow Volume Bias): Percent bias in the highest 2% of flows
- $PBIAS$ (Percent Bias): Overall percent bias across all flows

The general form for these biases is:

$$\text{PBIAS}_{\text{regime}} = \frac{\sum (y_{\text{pred,regime}} - y_{\text{target,regime}})}{\sum y_{\text{target,regime}}} \times 100\% \tag{37}$$

### E.2.5 KLING-GUPTA EFFICIENCY (KGE)

KGE combines correlation, bias, and variability components:

$$\text{KGE} = 1 - \sqrt{(r-1)^2 + \left(\frac{\sigma_{\text{pred}}}{\sigma_{\text{target}}} - 1\right)^2 + \left(\frac{\mu_{\text{pred}}}{\mu_{\text{target}}} - 1\right)^2} \tag{38}$$

where $r$ is the correlation coefficient, $\sigma$ represents standard deviation, and $\mu$ represents the mean.

## E.3 METRIC AGGREGATION

For each evaluation scenario (Random Holdout and Regional Holdout), metrics were calculated for each individual basin or station and then aggregated using median values to provide a robust measure of central tendency less sensitive to outliers. All metrics shown in tables throughout the paper represent these median values across the test set.

## E.4 IMPLEMENTATION DETAILS

All metrics were implemented in Python using NumPy for numerical computations and SciPy's statistical functions for correlation coefficients. Special care was taken to handle missing values (NaNs) appropriately in all calculations. For time series with missing values, only timestamps where both predicted and target values were available were used in metric calculations.

