# OpenReview forum: "StefaLand: An Efficient Geoscience Foundation Model That Improves Dynamic Land–Surface Predictions"
_ICLR.cc/2026/Conference — Submitted to ICLR 2026_

### Official Review · Reviewer_6uDU · 2025-10-17

**Soundness:** 2
**Presentation:** 1
**Contribution:** 1
**Rating:** 4
**Confidence:** 4

**Summary:**

This paper introduces StefaLand, a model that applies self-supervised learning with a Masked Autoencoder (MAE) to learn joint representations from both static landscape attributes and dynamic time-series geospatial data. The learned representations, pretrained using a cross-variable group masking strategy, are then fine-tuned to improve performance and spatial generalization on various downstream geoscience tasks.

**Strengths:**

This paper addresses two key limitations in current approaches to land surface process modeling: (1) existing supervised learning methods are prone to overfitting and exhibit poor generalization, and (2) existing self-supervised feature extraction methods trained on remote sensing imagery cannot be directly applied to land surface process modeling. To tackle these issues, the authors propose a self-supervised feature extraction method for remote sensing time-series data based on feature engineering combined with Masked AutoEncoder (MAE). They further demonstrate across multiple datasets that a self-supervised pretraining followed by fine-tuning strategy achieves better generalization than direct supervised learning.

**Weaknesses:**

1. The paper does not pose any scientifically meaningful question and reads essentially like a technical report. The critique of existing methods is superficial—limited to statements such as “poor performance” or “weak generalization”—without identifying the fundamental limitations of current self-supervised feature extraction methods specifically in the context of land surface process modeling. Moreover, the authors fail to discuss what concrete challenges arise when directly applying existing time-series self-supervised learning methods to this domain.


2. The paper provides a severely inadequate review of related work. Many of the land surface prediction tasks discussed are fundamentally time-series feature extraction and forecasting problems. However, the proposed “feature engineering + MAE” framework does not incorporate any modeling tailored to the physical processes or spatial dependencies inherent in land surface dynamics. In computer science, numerous self-supervised methods already exist for time-series and tabular data. The paper should clearly articulate how the proposed method differs from these established approaches and include experimental comparisons against them.


3. The experimental evaluation appears to deliberately understate baseline performance. For instance, Table 5 claims that baseline results come from:

Liu, J., Shen, C., Pei, T., Kifer, D., & Lawson, K. (2025). The value of terrain pattern, high‐resolution data and ensemble modeling for landslide susceptibility prediction. Journal of Geophysical Research: Machine Learning and Computation, 2, e2024JH000460. https://doi.org/10.1029/2024JH000460 .


However, upon reviewing this paper, I could not find the reported numbers: Logistic Regression ACC = 0.742 and Random Forest ACC = 0.751. Instead, the cited work reports Logistic Regression 1D ACC = 0.751, Random Forest 1D (RF1D) ACC = 0.821, CNN 2D (CNN2D) ACC = 0.880, and Random Forest 2D (RF2D) ACC = 0.765. Furthermore, several models in Liu et al. (2025) clearly outperform the proposed STEFALAND + CNN2D (ACC = 0.799). This casts serious doubt on the claimed performance improvements of STEFALAND.


4. The writing quality is poor. The main text lacks any flowcharts or visualizations of results. Additionally, the method section is confusing, conflating well-established techniques with the authors’ novel contributions without clear distinction.

**Questions:**

1. What is the core scientific problem addressed by this work, and what new insights does it offer to the research community?

2. Is the methodological design genuinely driven by the scientific problem? MAE was not originally developed for self-supervised representation learning on spatiotemporal geospatial data. What is the specific rationale for using MAE in land surface process modeling? Why were alternative self-supervised approaches—such as VAE, MoCo v3, SimCLR, or SCARF—not considered?

3. Compared to widely adopted self-supervised learning methods for time-series and tabular data in the machine learning community, what specific adaptations or optimizations does this work introduce for land surface process modeling? How would the proposed method perform in a fair comparison on standard, widely recognized benchmark datasets?

4. Can the authors provide detailed visual comparisons between their method and existing approaches?

---

> ### Author Response · Authors · 2025-11-22
>
> Weaknesses responses:
>
> 1. We agree the main points can be clarified better. We think three core scientific messages are already here but could be clarified in the abstract and discussion:
>    (1) capturing landscape organizational patterns can help dynamical prediction models generalize in space;
>    (2) improving data relevance, in addition to data size can elevate the supporting value of geoscientific foundation models.
>    (3) widely available scalar attribute data representations can efficiently provide critical information to geoscientific foundation models, and the previous emphasis on raw satellite images may not fit all tasks.
>    These points will be better reflected in the revised Research Questions. We would argue they are novel findings that could benefit the community and not necessarily promote a model per se.
> 2. We agree that the related work section can be strengthened. We added the new comparisons to the tables above and are working on additional foundational model comparisons. Our goal, however, is not to introduce a new architecture but to highlight what we see as the main scientific contributions we listed above. The broader geospatial foundation model landscape is difficult to navigate, and we did evaluate two existing Earth foundation models. This required substantial effort because of the data volume, preprocessing, and the fact that there were no prior demonstrations showing whether remote sensing or weather models would have any value for dynamical land surface prediction. We discussed this above in reply (2) to jPBF in relation to current available models.
>    You are correct that these tasks resemble standard time series feature extraction problems, but existing self supervised methods do not directly address the structure of land surface systems. These systems are driven by external forcings, a mix of static and dynamic attributes, and exhibit strong spatial organization that is not represented in generic MAE frameworks. We think it should be clear that our model has uniqueness given that it outperformed some well known foundation models in many tests (now AlphaEarth is added and we are working on one more). The idea of the residual adapter itself is not new, but using it in a CNN to LSTM pathway is a domain specific adaptation for this setting. These components are necessary for this domain and are not present in standard timeseries MAEs.
>    We will make these distinctions more explicit and clarify why existing self-supervised time series approaches cannot be applied directly without addressing these challenges.
>
> 3. Thank you for raising this point and a chance to clarify. In our initial experiments, we fine-tuned our StefaLand using data at a 30-m resolution (Liu et al., 2025, Figure 3). Thus, these baseline values were not directly comparable to the 10-m resolution results presented in Liu et al., 2025 Table 2\. To address this issue and ensure consistency, we have re-fine-tuned StafaLand using a 10-m resolution dataset. We have updated Table 5 accordingly, such that the baseline results now align directly with those reported by Liu et al. (2025). We notice this one has a distinctly higher recall.
>    Table 5
>    | Model  (10m)      | Accuracy ↑ | Precision ↑ | Recall ↑ | F1 ↑  | ROC AUC ↑ |
>    |--------------------|------------|-------------|----------|-------|-----------|
>    | Logistic Regression 2D | 0.744 | 0.720 | 0.795 | 0.756 | 0.823 |
>    | Random Forest 2D       | 0.765 | 0.737 | 0.822 | 0.777 | 0.849 |
>    | CNN2D                  | 0.880 | 0.896 | 0.858 | 0.877 | 0.854 |
>    | StefaLand \+ CNN2D      | 0.903 | 0.859 | 0.963 | 0.908 | 0.911 |
>    *Results show that StefaLand’s pretrained features improved the CNN’s generalization, yielding modest gains across most metrics. This is a particularly difficult baseline to improve, so even modest gains are rare. Recall increased from 0.858 to 0.963, and accuracy rose from 0.880 to 0.903, reflecting fewer misclassifications overall. While the random forest achieves high precision in this particular run, it does so at the cost of lower recall. Overall, StefaLand produces well-rounded predictions, with both higher accuracy and recall than CNN2D.*
>
> 4. We appreciate the feedback hopefully with added work and changes promised will alleviate this problem. We will include additional figures.

---

> > ### Author Response · Authors · 2025-11-22
> >
> > **Questions:**
> >
> > 1. Two messages discussed above.
> >
> > 2. The basic problem here is actually to capture **spatiotemporal organization** and **multi-domain process interactions** and how they jointly evolve. MAE is a natural candidate to enforce the model to capture the joint distribution by predicting missing information and has shown advantages in various tasks. VAE was attempted during the early developmental phase and did not result in good performance. The architecture of VAE makes it difficult to implement holdout masks to train on certain relationships.
> >    We will include this with the paper to clarify the choice: *“SimCLR and MoCO v3 are designed for images, but since we want to demonstrate that widely-available scalar attributes (rather than satellite images) can already provide valuable information, we avoided vision models in general. It is also extremely challenging for academic groups to train on massive satellite images with the limited resources at their disposal. SCARF is entirely for tabular data and does not meet our need of also incorporating temporal data. MAE was chosen as it is flexible enough to accommodate both scalar attributes and temporal data, with the potential of linking to features extracted from spatial fields. That said, any efficient architecture that allows us to capture spatiotemporal joint distribution could be worth examining, given available computing budgets."*
> >
> > 3. We partially addressed this above. However, to reiterate: in certain areas of hydrology like time series problems, the LSTM we provided demonstrates SOTA performance. The paper by Li et al. (2025) has extensive benchmarking on the CAMELS dataset that we used, including the two test cases for spatial generalization. Examining tables D2 through D3 in the appendix, you can see the results are roughly aligned. Do note they use an ensemble of models in three different flow regimes and take the mean performance. This testing practice tends to result in marginally better metrics. We also included the physics-based modeling that was done under the label dHBV in appendix table B1, but this is a slightly different problem structure involving parameterization of a physical model and so it was kept in the appendix.
> >    In the added tables we also include other self-supervised learning methods for time series and tabular data (Zeng et al. 2022; Zhou et al. 2021; Kitaev et al. 2020). Many available approaches focus on forecasting, which as we explain in our reply to jPBF is not commonly done in land surface modeling since these systems are input driven and, therefore, rely on forecasted forcings. We recognize that many of the methods used are not entirely new inventions. We combined several techniques along with careful dataset curation for pretraining to create a model setup that seems to offer particularly good performance in spatial generalization.
> >
> > 4. We're in the process of making the figure.
> >
> > We greatly appreciate your constructive inputs that helped us improve. We have added many new benchmark models, new results, and are working on more. We hope you see the uniqueness in our potential contribution to the community and if you have any more questions please let us know.
> >
> > References:
> >
> > Ailing Zeng, Muxi Chen, Lei Zhang, and Qiang Xu. *Are Transformers Effective for Time Series Forecasting?* arXiv preprint arXiv:2205.13504, 2022\.
> >
> > Haoyi Zhou, Shanghang Zhang, Jieqi Peng, Shuai Zhang, Jianxin Li, Hui Xiong, and Wancai Zhang. *Informer: Beyond Efficient Transformer for Long Sequence Time-Series Forecasting.* Proceedings of the Thirty Fifth AAAI Conference on Artificial Intelligence (AAAI 2021), volume 35, number 12, pages 11106 to 11115\. AAAI Press, 2021\.
> >
> > Nikita Kitaev, Lukasz Kaiser, and Anselm Levskaya. *Reformer: The Efficient Transformer.* International Conference on Learning Representations, 2020\.
> >
> > Li, P., Song, Y., Pan, M., Lawson, K., and Shen, C. Ensembling Differentiable Process-based and Data-driven Models with Diverse Meteorological Forcing Datasets to Advance Streamflow Simulation. EGUsphere preprint, [https://doi.org/10.5194/egusphere-2025-483](https://doi.org/10.5194/egusphere-2025-483)
> >
> > Liu, J., Shen, C., O'Donncha, F., Song, Y., Zhi, W., Beck, H. E., Bindas, T., Kraabel, N., and Lawson, K. From RNNs to Transformers: benchmarking deep learning architectures for hydrologic prediction. EGUsphere preprint, [https://doi.org/10.5194/egusphere-2025-1706](https://doi.org/10.5194/egusphere-2025-1706)
> >
> > Jiangtao Liu, Te Pei, Chaopeng Shen, , Daniel Kifer, and Kathryn Lawson. The value of terrain pattern, high-resolution data and ensemble modeling for landslide susceptibility prediction. ESS Open Archive preprint, June 2025\. URL http://dx.doi.org/10.22541/essoar.175130065.56738480/v1.

---

> > > ### Comment · Reviewer_6uDU · 2025-11-23
> > >
> > > My evaluation is based on whether the paper makes a methodological contribution to self‑supervised learning, beyond being a careful application of an existing paradigm to a new domain. From that perspective, the core problem here is self‑supervised learning on (spatio)temporal tabular data. When I mentioned VAE, SimCLR, SCARF, etc., I was referring to paradigms (generative/compression, contrastive, masked‑reconstruction), not specific image architectures, and all of these paradigms already have concrete instantiations on temporal/tabular data.
> > >
> > > In particular, there is substantial prior work on self‑supervised learning for tabular and temporal tabular data using corruption‑based or contrastive objectives (e.g., SCARF, SubTab, TS‑TCC, TS‑MoCo, temporal supervised contrastive learning for clinical trajectories) [1–5], and on BERT/MAE‑style masked modeling directly on sequential tabular data (e.g., TabBERT, UniTTab) [6,7]. These works show that capturing joint spatiotemporal structure of scalar attributes is not specific to MAE.
> > >
> > > The current response does not provide a mathematical or conceptual argument that MAE is necessary or uniquely appropriate for this problem, nor does the paper engage with prior masked‑reconstruction methods for temporal/tabular data (such as TabBERT/UniTTab), nor present experimental comparisons against representative alternatives from other self‑supervised paradigms. If the authors’ claim is that their approach advances self‑supervised learning, then what needs to be demonstrated is not only that the method works well, but that this particular masked‑autoencoding formulation is better suited to the problem than standard contrastive, variational, or alternative masked approaches.
> > >
> > > Absent such theoretical justification or comparative evidence, the work reads primarily as a technically solid application of an existing MAE‑style idea to an important geoscience problem, rather than a contribution that clearly advances the methodology of self‑supervised learning in the sense expected for ICLR.
> > >
> > > [1] Labonne et al., SCARF: Self-Supervised Contrastive Learning using Random Feature Corruption for Tabular Data, 2021.
> > > [2] Ucar et al., SubTab: Subsetting Features of Tabular Data for Self-Supervised Representation Learning, 2021.
> > > [3] Eldele et al., Time-Series Representation Learning via Temporal and Contextual Contrasting, 2021.
> > > [4] Bahrami et al., TS‑MoCo: Time-Series Momentum Contrast for Self-Supervised Physiological Representation Learning, 2023.
> > > [5] Zhang et al., Temporal Supervised Contrastive Learning for Modeling Patient Risk Progression, 2023.
> > > [6] Somepalli et al., Tabular Transformers for Modeling Multivariate Time Series (TabBERT), 2020.
> > > [7] Zerveas et al., One Transformer for All Time Series: Representing and Training with Time-Dependent Heterogeneous Tabular Data (UniTTab), 2025.

---

> > > > ### Comment · Reviewer_UBGy · 2025-11-23
> > > > **I think this is within scope**
> > > >
> > > > Dear Reviewer 6uDU,
> > > >
> > > > according to the ICLR CfP https://iclr.cc/Conferences/2026/CallForPapers :
> > > > > A non-exhaustive list of relevant topics:
> > > > > - [...]
> > > > > - applications to physical sciences (physics, chemistry, biology, etc.)
> > > >
> > > > This work represents an important application for land surface hydrology, which is a physical science. The method used is novel in that application field. Therefore I do believe this work is within scope for ICLR.
> > > >
> > > > Kindly,
> > > > Reviewer UBGy

---

> > > > > ### Comment · Reviewer_6uDU · 2025-11-24
> > > > >
> > > > > Dear Reviewer UBGy,
> > > > >
> > > > > thank you very much for your clarification. I did not mean to suggest that the work is outside the AI4Science scope, but rather to emphasize that the “AI” and “Science” aspects are equally important in the evaluation.
> > > > >
> > > > > In my view, simply applying a well-studied method from computer science to a scientific domain does not automatically constitute a new methodological contribution. Instead, it is important to demonstrate that the method is adapted to the specific characteristics of the target scientific domain. For example, in geoscience problems, issues such as spatial autocorrelation and spatio-temporal heterogeneity should be explicitly addressed.
> > > > >
> > > > > Concretely, I believe a strong AI4Science paper should:
> > > > >
> > > > > Systematically analyze why existing methods are insufficient for the scientific problem at hand (here, this would include a theoretical discussion of why current tabular self-supervised learning methods struggle with spatio-temporal, multi-source heterogeneous data);
> > > > >
> > > > > Propose reasonable, domain-informed modifications that explicitly leverage the characteristics of the scientific domain (in this case, how the method is specifically adapted to spatio-temporal structure and multi-source heterogeneity in geospatial data);
> > > > >
> > > > > Provide systematic evidence that these adaptations are indeed necessary, i.e., that one cannot simply reuse off-the-shelf methods from the CS literature to obtain comparable performance (for this paper, this would mean demonstrating that a MAE tailored to geospatial data substantially outperforms existing self-supervised approaches designed for generic spatio-temporal tabular data).
> > > > >
> > > > > I hope this clarifies my original intention, and I appreciate your helpful remarks.

---

> > > > > > ### Author Response · Authors · 2025-11-25
> > > > > >
> > > > > > Thank you again for your detailed and insightful review. We now recognize that your evaluation is framed around whether the paper provides a methodological contribution to self-supervised learning (SSL). Your comments helped us understand that our framing may have unintentionally suggested a broader SSL contribution than intended. We appreciate the opportunity to clarify and to revise the manuscript so that its scientific contribution is communicated more clearly.
> > > > > >
> > > > > > 1\. On alternative SSL paradigms
> > > > > > We thank the reviewer for the seasoned and technically informed suggestions regarding generative, contrastive, and masked paradigms. These remarks help situate our work within the broader SSL landscape, and we appreciate the opportunity to refine this part of the paper.
> > > > > > While these directions are valuable, our evaluation follows established practice in large Earth foundation models such as Prithvi, Terramind, AlphaEarth, and Aurora (Bodnar et al., Nature, 2025), which also employ a single SSL paradigm. This is due in part to the massive data volumes and the combinatorial space of pretraining strategies, adapters, finetuning configurations, and dataset choices. These models likewise do not conduct full cross-paradigm comparisons, and our study aligns with this evaluation norm.
> > > > > > Furthermore, existing SSL approaches for tabular or time-series data (SCARF, SubTab, TS-TCC, TabBERT, UniTTab) operate under assumptions that do not directly accommodate heterogeneous, multi-source geospatial data:
> > > > > > 	•	features are treated as homogeneous;
> > > > > > 	•	static and dynamic inputs are not jointly modeled;
> > > > > > 	•	masking is typically uniform across timesteps or variables.
> > > > > > Our model instead supports variable-specific and time-specific random masking, which enables patterns such as masking only selected variables over selected temporal segments—functionality not present in standard MAE/tabular SSL frameworks.
> > > > > > We will expand our discussion to contrast our design choices with these paradigms and highlight how their principles might inspire future tailored SSL approaches for geoscience. For example, we will include the following:
> > > > > > “While SSL paradigms such as SCARF, SubTab, TS-TCC, TabBERT, and UniTTab are promising, their current formulations assume homogeneous features and uniform masking and therefore do not support the variable-wise and time-wise random masking required for heterogeneous geospatial inputs. Their underlying ideas may, however, inspire future extensions tailored to land-surface systems.”
> > > > > >
> > > > > > 2\. Nature of the contribution
> > > > > > We do not claim to introduce a new SSL paradigm. Rather, the main contribution is scientific and insight-focused, consistent with AI4Science objectives:
> > > > > > 	•	demonstrating that curated scalar attributes can efficiently encode cross-domain process interactions, providing a lightweight, interpretable alternative to raw imagery for downstream land-surface tasks;
> > > > > > 	•	demonstrating that curated multi-source scalar attributes+time series MAE can form an effective pretraining channel;
> > > > > > 	•	showing that these domain-informed attributes measurably improve spatial generalization, a long-standing challenge in hydrology and land-surface modeling;
> > > > > > 	•	evaluating against several geoscience foundation models (Prithvi, Terramind, AlphaEarth) that use very different pretraining strategies and much larger data volumes;
> > > > > > 	•	and demonstrating applicability across hydrology, soil applications, and landslides, spanning multiple major geoscience subfields with large communities.
> > > > > > These findings relate to the scientific structure of land-surface systems and may also be relevant for domains where meaningful scalar and temporal attributes (rather than raw imagery) are available.
> > > > > > We will revise the introduction and contribution statement to foreground these scientific questions more explicitly, as the reviewer suggested.

---

> ### Author Response · Authors · 2025-11-25
>
> 3\. Domain-specific architectural adaptations
> Although our approach draws on MAE concepts, several elements were designed specifically for geoscience:
> 	•	Variable- and time-specific masking: different variables can be masked across different temporal segments, e.g., masking rainfall while unmask temperature on the same day, enabling the model to learn cross-variable temporal interactions;
> 	•	Residual adapter enables static spatial attributes to fused with the LSTM temporal pathway, a design validated in ablations;
> 	•	Coupled static–dynamic training objective: reflecting that land-surface processes are driven jointly by landscape characteristics and external forcings.
> We agree that these points should be described more clearly in the paper, and we will revise Section 3 accordingly.
>
> 4\. Reframing the expected contribution
> We appreciate the reviewer’s emphasis that strong AI4Science work should integrate scientific insight with appropriate machine-learning methodology. With this framing:
> 	•	we do not claim MAE is uniquely optimal;
> 	•	we do not propose a new SSL paradigm;
> 	•	instead, the work advances understanding of how static–dynamic interactions govern spatial generalization;
> 	•	and provides a tailored SSL configuration that operationalizes this insight.
> Your feedback has helped us significantly improve how we communicate this distinction. We will update the manuscript to ensure the scientific scope and methodological choices are presented with much greater clarity.
> This work represents a broadly applicable contribution within geoscience AI, demonstrated across four diverse land-surface tasks. We sincerely appreciate the reviewer’s rigorous and thoughtful feedback, which has strengthened the paper. We hope that the clarified framing will help evaluate the work within its intended scientific context.
>
> Bodnar et al., Nature, A foundation model for the Earth system, doi: 10.1038/s41586-025-09005-y

---

### Official Review · Reviewer_UBGy · 2025-11-01

**Soundness:** 2
**Presentation:** 2
**Contribution:** 4
**Rating:** 4
**Confidence:** 5

**Summary:**

This work introduces a foundation model for land surface hydrology. The model is a transformer that is pre-trained using a masked auto-encoder to reconstruct co-located meteorlogy time series and static environmental characteristics related to vegetation cover, soil properties, topography and human influence. It is evaluated on four downstream tasks: streamflow prediction, landslide susceptibility mapping, soil moisture prediction and sand/clay-fraction prediction. It is comparable or outperforms supervised baselines on these tasks, with particularly promising benefits for streamflow prediction.

**Strengths:**

1. Focus on hydrology, which can have important impacts on societies resilience to extreme events, by improving early warning systems
2. Large benefits on streamflow prediction from using the pre-trained transformer backbone.
3. Competitive performance on four tasks related to land surface hydrology
4. Evaluation of spatial extrapolation and honest reporting of a drop in performance (akin to other supervised baselines) in the challenging regional holdout scenario
5. Interesting additional experiment to leverage the embeddings for predicting parameters of a hydrological model.

**Weaknesses:**

Major points:

1. No previously published baselines included in the downstream task evals (except for the landslide task). This is a major point, as it is hard to assess how well the StefaLand model performs against current SOTA. The authors do make an effort to include commonly used models as baselines, however it still remains unclear how competitive StefaLand really is.
2. Soil properties downstream task suffers from potentially skewed evaluation: Prediction the soil properties (albeit from a different harmonized dataset) is one of the pre-training task for StefaLand.
3. Presentation lacks clarity at times:
   -  L141 x & b not introduced
   - L189 it is confusing that the exact groupings are not mentionend here, and it is also not elucidated how the "reciprocal or bidirectional causality" is determined
   - Section 2.3 : Figure D1 definitely indicates a different structure than Equations 13-15. But i'd argue that actually both require reworking: for each operation / neural network module it needs to be clear what the inputs and outputs are, especially, how embeddings, static variables and forcings are routed. Also: what happens with predictors that have not been part of the pretraining.
   - Table 1 (and the following) should probably be re-ordered for more clarity. Currently it combines two aspects: 1) Performance scores against baselines on CAMELS and 2) ablation studies. For clarity, these two should be separate. So the message fro Table 1 can be: StefaLand gives performance benefits over baseliens on CAMELS. And the additonal tables' message is: the best decoder strategy for stefaland is to use a resconn decoder. Then, the order of rows shoudl also be changed, such that baselines are on top and the "chosen" model is on the bottom.
   - Paragraph L282-L288 is confusing. It does not mention the nuances between the different papers, reads more like a related work section, and raises the question why these baselines were not included in the eval, but instead the authors took it on themselves to train a supervised LSTM.
   - throughout the manuscript the same topics re-appear multiple times. It would be really good if the manuscript would be overall streamlined again into a common storyline, and less of a listing of different things that were done and justifications for design choices.
4. Experiments are a bit strange. Typically for a foundation model, i'd expect the following experiments: training from scratch, zero-shot (if possible), linear probing and fine-tuning. However, this work only reports training from scratch and fine-tuning, but does so with multiple different decoders (see comment above). Again, i'd suggest to change the narrative here also such that its easier to understand for the community: In table 1 include the baselines LSTM, TerraMind & ideally the SOTA from literature, and then include StefaLand from scratch (with resconn decoder), linear probing (just linear layer...) & fine-tuning (with resconn). This allows much more to understand the value of the pre-trained embeddings vs. the model architecture. Then put the ablation of different decoders in a different table. Ideally use as metric there not just 1 downstream task, but (avg) performance across all four downstream tasks, to showcase that you are proposing a robust model architecture for the foundation model. By the way,  I think for soil property prediction you can include zero-shot prediction.
5. The other foundation models included in this study make little sense for the downstream tasks (except for terramind for soil mapping - but its not evaluated there). It is obvious that TerraMind and PrithviWxC are not made for streamflow or soil moisture prediction. So if you insist on keeping them in there, perhaps try to tone down the narrative throughout the manuscript a bit - you have enough interesting content, so there is really no need to emphasize so much on this comparison.
6. This works framing seems a bit to broad. The four downstream tasks are all land surface hydrology tasks, and in that context they are tasks that perhaps depend less on the the interannual variability of the vegetation, thus its also unclear how well the approach with the static features would work for land surface tasks beyond hydrology (e.g. carbon, energy ,...). I suggest to dampen the framing a bit and explicitly frame this work as a foundation model for land surface hydrology -- which by the way is still super impactful.


Minor points:

7. Which hyperparameters did you tune?
8. Where is the ablation of the different adapters in Figure D3?
9. L052 "represents the impacts of climate change" seems a bit of a strong statement and to my understanding not entirely true.
10. Tables comparing against supervised baselines should include a comparison of parameter numbers and training / inference cost -- as i understand that these are not necessarily always fair comparisons in these terms.
11. not sure if your downstream tasks really always use the most predictive inputs.. e.g. you seem to neglect remote sensing for soil properties and moisture prediction...

**Questions:**

An analysis of the different pre-training objectives could actually be interesting. How good were your reconstruction losses? Which variable groups were predictive for which others? etc. :-)

---

> ### Author Response · Authors · 2025-11-22
>
> Weaknesses response:
>
> 1. We apologize if our paper was not written clearly. We understand it is difficult for the ML community to believe LSTM as the SOTA, but LSTM is very difficult to beat in this case and many papers with rigorous evaluations acknowledge this (Wang et al., 2024), (Waqas and Humphries, 2024), (Kratzert et al., 2024), (Liu et al., 2025). Some papers find a new network with slightly better results on a different (typically even smaller) dataset. Also, we now add Dlinear, Informer and reformer as comparisons in the temporal tests (Table 1 and 3 above) and as you can see their results are worse. This is due to the fact that LSTM fits the nature of the problem very well.
>
> 2. Thanks for the point. Not surprising that one thinks so but there is a large discrepancy between the two datasets and this is a large part of the reason for demonstrating this task. We are trying to reconcile these two datasets. In the pretraining the soil properties from HWSD are used as an input, but for this task they are also an input. There is no data leakage to the target because the ISRIC values we are predicting are not seen during pretraining.
>
> 3. Presentation lacks clarity at times:
>    1. X is the per variable input and b is the learnable bias. We will add a sentence to that section.
>    2. Good point. We will make these groupings clearer, but in pretraining predictors are grouped into broad categories, including climate (precipitation, short-wave radiation, humidity, temperature, and potential evapotranspiration), vegetation (for example forest and grassland cover fractions from CCI land cover and NDVI), soil properties (for example sand, silt, and clay fractions), topography (for example elevation, slope, aspect, and soil depth), geology (for example carbonate sedimentary rock fraction, porosity, and permeability), and socioeconomic attributes (such as population density or GDP per capita). The groupings were meant to be straightforward and we should note they do not strictly follow the statistical definition of bidirectional causality. Pairs such as precipitation and potential evapotranspiration can have strong statistical associations without meeting the formal bidirectional criteria.
>
>    C-F We reworked the tables given your feedback and can reword some of the methodology to make it clear what we did. The methodology section primarily addresses the foundation model and we will add an appendix section on the adapters tested with the ablations for the differing designs tested and sizes included.
>
>
> 4. Good points: we have revised the tables and they were included in the post (jPBF response comment 1). We do not  believe any of the tasks can be done zero-shot.
>
> 5. We will make a revision to tone down the manuscript and do our best to add some more comparisons to other models. We do not intend for this to be an attack on these two models. The message is that data relevance using scalar attributes is of critical importance and it may not be that the more data we include in pretraining the better, and the advanced vision transformer architecture may not be the right choice for some tasks.
>
> 6. We appreciate the comment. The soils and landslide tasks are not only related to hydrology and are typically not regarded as the hydrologic domain. These papers are generally published in journals outside of hydrology domain journals. Carbon and energy cycles are part of potential future work and the pretraining tasks have shown promise for the model to support ecosystem and carbon flux predictions. Our preliminary results on this front are already showing a huge potential for carbon cycle variables but it is too much to  include in this manuscript. We will seek to moderately tone down the scope. However, we also understand there are other foundation models that could be of significant value as well, and will adjust the message accordingly.
>
> Minor points:
>
> 7. We tuned the hyperparameters learning rate, hidden sizes, encoding and decoding layer counts. Along with testing optimizers Adam, AdamW and Adadelta.
>
> 8. We will add these to the appendix
>
> 9. We will remove this claim.
>
> 10. There is some limited analysis of comparison of model cost in terms of time per epoch in the appendix. However, we can expand this.
>
> 11. Is there a particular input or forcing you think is missing here that is concerning? We used 16 different dynamic input forcings for the soil moisture case, including MODIS albedo and land surface temperature, GPM and MSWEP precipitation, and the full ERA5 surface and vegetation fields. The static inputs also already include MODIS and CCI land cover and NDVI. For the soil composition task we use the same MSWX and MSWEP forcings along with a full stack of RS derived static attributes.

---

> > ### Author Response · Authors · 2025-11-22
> >
> > **Questions:**
> >
> > Thank you. We will work on further analysis. We have provided quantification of errors for the reconstruction loss through inference; you can view them below.
> >
> > Time Varied
> >
> > | Variable | R² | L1 | L2 |
> > |---------|------|---------|---------|
> > | P | 0.970 | 0.330 | 1.252 |
> > | SWd | 0.938 | 11.891 | 20.212 |
> > | RelHum | 0.955 | 1.589 | 3.423 |
> > | Tmax | 0.958 | 1.398 | 3.155 |
> > | Tmin | 0.979 | 1.251 | 3.019 |
> >
> > Static
> > | Variable | L1 | L2 |
> > |----------|---------|---------|
> > | meanP | 15.408 | 56.618 |
> > | seasonality P | 0.232 | 0.281 |
> > | seasonality PET | 0.216 | 0.266 |
> > | snow\_fraction | 0.176 | 0.207 |
> > | snowfall\_fraction | 0.120 | 0.143 |
> > | meanTa | 7.376 | 9.467 |
> > | mean elevation | 401.029 | 606.122 |
> > | meanslope | 4.682 | 5.636 |
> > | aspectcosine | 0.206 | 0.268 |
> > | HWSD\_clay | 9.085 | 12.074 |
> > | HWSD\_sand | 15.590 | 18.798 |
> > | HWSD\_silt | 10.980 | 13.082 |
> > | porosity | 0.062 | 0.072 |
> > | Permeability\_no\_permafrost | 0.895 | 1.098 |
> > | carbonate\_sedimentary\_rocks\_frac | 0.189 | 0.264 |
> > | soil\_depth | 10.530 | 14.215 |
> > | NDVI | 0.137 | 0.171 |
> > | grassland\_fraction\_2018 | 0.212 | 0.276 |
> > | forest\_fraction\_1992 | 0.335 | 0.372 |
> > | forest\_fraction\_2018 | 0.329 | 0.371 |
> > | population\_density\_2000\_buffer | 78.363 | 123.232 |
> > | gdp\_2000\_buffer | 125.181 | 222.406 |
> > | intact\_forest | 0.000 | 0.000 |
> >
> > Thank you for your valuable comments. In summary, we added many models for comparison (still working on more), added AlphaEarth, and actually found better results to support our arguments. Please kindly let us know if you have more questions.
> >
> > References:
> > Wang, Y., Shi, L., Hu, Y., Hu, X., Song, W., and Wang, L. A comprehensive study of deep learning for soil moisture prediction. *Hydrology and Earth System Sciences*, 28(4), 917 to 943, 2024\. [https://doi.org/10.5194/hess-28-917-2024](https://doi.org/10.5194/hess-28-917-2024)
> >
> > Waqas, M., and Humphries, U. W. A critical review of RNN and LSTM variants in hydrological time series predictions. *MethodsX*, 13, 102946, 2024\. [https://doi.org/10.1016/j.mex.2024.102946](https://doi.org/10.1016/j.mex.2024.102946)
> >
> > Kratzert, F., Gauch, M., Klotz, D., and Nearing, G. HESS Opinions: Never train a Long
> > Short-Term Memory (LSTM) network on a single basin. *Hydrology and Earth System Sciences*, 28(17), 4187 to 4201, 2024\. [https://doi.org/10.5194/hess-28-4187-2024](https://doi.org/10.5194/hess-28-4187-2024)
> >
> > Liu, J., Shen, C., O'Donncha, F., Song, Y., Zhi, W., Beck, H. E., Bindas, T., Kraabel, N., and Lawson, K. From RNNs to Transformers: benchmarking deep learning architectures for hydrologic prediction. *EGUsphere* (preprint), 2025\. [https://doi.org/10.5194/egusphere-2025-1706](https://doi.org/10.5194/egusphere-2025-1706)

---

> > > ### Comment · Reviewer_UBGy · 2025-11-23
> > >
> > > Dear authors, thanks a lot for the rebuttal. I am willing to raise my score if you address this last point:
> > >
> > > > We apologize if our paper was not written clearly. We understand it is difficult for the ML community to believe LSTM as the SOTA, but LSTM is very difficult to beat in this case and many papers with rigorous evaluations acknowledge this (Wang et al., 2024), (Waqas and Humphries, 2024), (Kratzert et al., 2024), (Liu et al., 2025). Some papers find a new network with slightly better results on a different (typically even smaller) dataset. Also, we now add Dlinear, Informer and reformer as comparisons in the temporal tests (Table 1 and 3 above) and as you can see their results are worse. This is due to the fact that LSTM fits the nature of the problem very well.
> > >
> > > My point is merely, you should include numbers from previously published papers. It is unclear if the LSTM you trained here is really as competitive as the one used by Google (or whoever else is currently SOTA), because you compare on different datasets and do not directly compare in the same evaluation setting (and do not include previously published SOTA numbers).
> > > This, by the way, is also super important for the field as a whole -- if everyone evaluates their models on their own evaluation datasets, it becomes impossible to really track progress.

---

> ### Author Response · Authors · 2025-11-25
>
> We completely understand this concern. This is a known issue in hydrology evaluation practice. To address it, we assembled tables that place our results next to previously published numbers wherever the evaluation setting was directly compatible. In two cases we had to reproduce the published setup ourselves, since many hydrology papers use different sequence lengths or windowing conventions. We will point this out explicitly. It is also common that papers report only a headline metric, which makes it hard to compare subtle differences, and many studies use multi model ensembles that improve scores.
>
> We also want to note that the LSTM we used is, for all practical purposes, the same core model class implemented in the main hydrology papers that use an LSTM in hydrologic prediction (Feng et al., 2023; Sabzipour et al., 2023; Liu et al., 2025). There are differences in hyperparameter tuning, training targets, and windowing choices, but the underlying network definition does not differ in any meaningful way.
>
> This is from “Ensembling Differentiable Process-based and Data-driven Models with Diverse Meteorological Forcing Datasets to Advance Streamflow Simulation” table D2-3 of the appendix of this paper. The evaluation setup matches ours: same CAMELS basins, same random and regional holdouts, same sequence-to-sequence prediction structure. The main difference is that their reported ensemble scores come from training three separate models for three flow regimes and averaging the outputs. This involves substantially more total training and typically yields boosted performance. We include their single-model LSTM, their δHBV hybrid, their ensemble variant, and our reproduced values under the same conditions.
>
> Random Holdout
>
> | Source | Model                             | NSE      | RMSE    |
> |--------|------------------------------------|----------|---------|
> | Paper  | LSTM                               | 0.702636 | 1.31714 |
> | Paper  | (LSTM Parameterized HBV) δHBV      | 0.706809 | 1.35541 |
> | Paper  | Ensembled LSTM+δHBV                | 0.74227  | 1.24887 |
> | Ours   | LSTM                               | 0.636    | 1.402   |
> | Ours   | StefaLand-ResConn                  | 0.717    | 1.111   |
> | Ours   | StefaLand Parameterized HBV        | 0.714    | 1.234   |
>
> Regional Holdouts
>
> | Source | Model                             | NSE      | RMSE    |
> |--------|------------------------------------|----------|---------|
> | Paper  | LSTM                               | 0.578365 | 1.59111 |
> | Paper  | (LSTM Parameterized HBV) δHBV      | 0.622278 | 1.57189 |
> | Paper  | Ensembled LSTM+δHBV                | 0.644398 | 1.46185 |
> | Ours   | LSTM                               | 0.554    | 1.609   |
> | Ours   | StefaLand-ResConn                  | 0.635    | 1.344   |
> | Ours   | StefaLand Parameterized HBV        | 0.643    | 1.345   |
>
> These results are from “Toward Improved Predictions in Ungauged Basins: Exploiting the Power of Machine Learning” table 2\. Note Global LSTMs are not testing on the same framework; these were included to show how much performance decreases when basins are unseen.  This paper used 12 random holdouts but the same underlying Camels data. We reran their experiment to get comparable results.
>
> | Source | Model                            | NSE   |
> |--------|----------------------------------|-------|
> | Paper  | PUB LSTM                         | 0.69  |
> | Paper  | SAC-SMA                          | 0.64  |
> | Paper  | NWM                              | 0.58  |
> | Ours   | LSTM                             | 0.678 |
> | Ours   | StefaLand-ResConn                | 0.728 |
> | Easier Temporal Test Regime |
> | Paper  | *Global LSTM (no statics)*       | 0.63  |
> | Paper  | *Global LSTM (with statics)*     | 0.74  |

---

> > ### Author Response · Authors · 2025-11-25
> >
> > Spatial Cross-validation from “From RNNs to Transformers: benchmarking deep learning architectures for hydrologic prediction.” Table S5. Note it is not clear that this is the exact same spatial cross validation we ran some details are missing from this table about what it is referring too in terms of cross validation testing on camels however based on some context clues within the paper I believe this is the same testing setup we used as it is referencing the same paper we are basing this testing scheme off of. Their LSTM was trained with an NSE loss, while our implementation uses RMSE, which changes optimization dynamics but not the underlying model structure. We report their published values alongside our reproduced results under the equivalent spatial split.
> >
> > | Source | Model | NSE | ubRMSE |
> > |--------|------------------------------|-------|--------|
> > | Paper | DLinear | 0.33 | 2.01 |
> > | Paper | LSTM | 0.67 | 1.38 |
> > | Paper | Informer | 0.60 | 1.48 |
> > | Paper | Reformer | 0.61 | 1.49 |
> > | Paper | iTransformer | 0.34 | 1.97 |
> > | Paper | Transformer | 0.61 | 1.43 |
> > | Paper | ETSformer | 0.62 | 1.47 |
> > | Paper | Pyraformer | 0.64 | 1.41 |
> > | Paper | Crossformer | 0.62 | 1.49 |
> > | Paper | CARDformer | 0.29 | 2.05 |
> > | Paper | Non-stationary Transformer | 0.61 | 1.49 |
> > | Ours | LSTM | 0.636 | 1.36 |
> > | Ours | StefaLand-ResConn | 0.717 | 1.068 |
> >
> > From paper “Evaluating a global soil moisture dataset from a multitask model (GSM3 v1.0) with potential applications for crop threats” Table 1\. These are the same test setups presented here as we ran.
> >
> > Random Holdout
> >
> > | Source | Model | RMSE | ubRMSE | Corr |
> > |--------|--------------------|-------|--------|-------|
> > | Paper | Multi-task LSTM | 0.075 | 0.056 | 0.792 |
> > | Our | LSTM | 0.073 | 0.055 | 0.764 |
> > | Our | StefaLand-ResConn | 0.068 | 0.054 | 0.783 |
> >
> > Regional Holdout (Europe)
> >
> > | Source | Model | RMSE | ubRMSE | Corr |
> > |--------|--------------------|-------|--------|-------|
> > | Paper | Multi-task LSTM | 0.104 | 0.062 | 0.646 |
> > | Our | LSTM | 0.112 | 0.053 | 0.510 |
> > | Our | StefaLand-ResConn | 0.090 | 0.059 | 0.638 |
> >
> > This is from the paper “ A comprehensive study of deep learning for soil moisture prediction.” Figure 9-11. The authors do not specify which ten sites were used in their smaller subset, so only the thirty site configuration can be matched. Because the paper reports these results only in plot form, we extract approximate RMSE values from the figures for the one day lead time at 0.05 m, 0.10 m, and 0.20 m depths. These extracted values are approximate and are provided for context, since they show that the relative ranking of models is consistent across datasets and depth dependent benchmark
> >
> > | Source | Model | Depth | RMSE |
> > |--------|-------------|--------|-------|
> > | Paper | CNN | 0.05m | \~1.4 |
> > | Paper | CNN | 0.10m | \~1.2 |
> > | Paper | CNN | 0.20m | \~0.8 |
> > | Paper | Transformer | 0.05m | \~1.41 |
> > | Paper | Transformer | 0.10m | \~1.21 |
> > | Paper | Transformer | 0.20m | \~0.84 |
> > | Paper | LSTM | 0.05m | \~1.35 |
> > | Paper | LSTM | 0.10m | \~1.15 |
> > | Paper | LSTM | 0.20m | \~0.75 |
> > | Paper | FA-LSTM | 0.05m | \~1.35 |
> > | Paper | FA-LSTM | 0.10m | \~0.98 |
> > | Paper | FA-LSTM | 0.20m | \~0.70 |
> > | Our | LSTM | 0.05m | 1.381 |
> > | Our | LSTM | 0.10m | 1.127 |
> > | Our | LSTM | 0.20m | 0.83 |
> > | Our | StefaLand-ResConn | 0.05m | 1.431 |
> > | Our | StefaLand-ResConn | 0.10m | 1.311 |
> > | Our | StefaLand-ResConn | 0.20m | 0.812 |

---

> > > ### Author Response · Authors · 2025-11-25
> > >
> > > References:
> > >
> > > Feng, D., Beck, H., Lawson, K., and Shen, C. The suitability of differentiable, physics-informed machine learning hydrologic models for ungauged regions and climate change impact assessment. _Hydrology and Earth System Sciences_, 27, 2357 to 2373, 2023\. https://doi.org/10.5194/hess-27-2357-2023
> > >
> > > Sabzipour, B., Arsenault, R., Troin, M., Martel, J., Brissette, F., Brunet, F., and Mai, J. Comparing a long short-term memory neural network with a physically based hydrological model for streamflow forecasting over a Canadian catchment. _Journal of Hydrology_, 627, 130380, 2023\. [https://doi.org/10.1016/j.jhydrol.2023.130380](https://doi.org/10.1016/j.jhydrol.2023.130380)
> > >
> > > Wang, Y., Shi, L., Hu, Y., Hu, X., Song, W., and Wang, L. A comprehensive study of deep learning for soil moisture prediction. _Hydrology and Earth System Sciences_, 28(4), 917 to 943, 2024\. [https://doi.org/10.5194/hess-28-917-2024](https://doi.org/10.5194/hess-28-917-2024)
> > >
> > > Liu, J., Shen, C., O'Donncha, F., Song, Y., Zhi, W., Beck, H. E., Bindas, T., Kraabel, N., and Lawson, K. From RNNs to Transformers: benchmarking deep learning architectures for hydrologic prediction. _EGUsphere_ (preprint), 2025\. [https://doi.org/10.5194/egusphere-2025-1706](https://doi.org/10.5194/egusphere-2025-1706)
> > >
> > > Li, P., Song, Y., Pan, M., Lawson, K., and Shen, C. Ensembling Differentiable Process-based and Data-driven Models with Diverse Meteorological Forcing Datasets to Advance Streamflow Simulation. EGUsphere preprint, [https://doi.org/10.5194/egusphere-2025-483](https://doi.org/10.5194/egusphere-2025-483)
> > >
> > > Kratzert, F., Klotz, D., Herrnegger, M., Sampson, A. K., Hochreiter, S., & Nearing, G. S. (2019). Toward improved predictions in ungauged basins: Exploiting the power of machine learning. _Water Resources Research_, 55, 11344–11354. [https://doi.org/10.1029/2019WR026065](https://doi.org/10.1029/2019WR026065)
> > >
> > > Jiangtao Liu, David Hughes, Farzad Rahmani, Kathryn Lawson, and Chaopeng Shen. Evaluating a global soil moisture misc from a multitask deep learning model. Geoscientific Model Development, 16(5):1553–1567, 2023b. doi:10.5194/gmd-16-1553-2023

---

### Official Review · Reviewer_jPBF · 2025-11-02

**Soundness:** 2
**Presentation:** 2
**Contribution:** 2
**Rating:** 2
**Confidence:** 5

**Summary:**

This paper proposes StefaLand, a foundation model designed for land surface monitoring tasks. The model is a transformer encoder that takes in point-based covariates (attribute-formatted rather than pixel/raster-formatted) from static and time-varying data sources such as precipitation, humidity, forest cover fraction, soil depth, population density, and more. Experiments on five datasets shows that StefaLand outperforms the supervised baseline and use of PrithviWxC and TerraMind features in place of StefaLand features. The model is much smaller than foundation models designed for raster datasets, making it more accessible for researchers than adapting larger vision foundation models.

**Strengths:**

- StefaLand is much smaller than foundation models designed for raster datasets, making it more accessible for researchers than adapting larger vision foundation models.
- The paper aims to address land surface modeling tasks that involve forecasting dynamic, time-varying phenomena like soil moisture, streamflow runoff, etc. This is different than many other geospatial foundation model papers that focus on land surface detection or mapping tasks like land cover classification.
- StefaLand has better performance than the supervised baseline and PrithviWxC or TerraMind features on all tasks.

**Weaknesses:**

- The Intro says that the goal of StefaLand is to “predict what will happen in the near or distant future”, but it does not seem like any of the downstream tasks are forecasting future timesteps. Maybe I don’t understand these tasks, but are the tasks predicting *future* states based on previous timesteps, like future soil characteristics or streamflow?
- The paper says they “could not identify other earth foundation models that are designed for land-surface predictions”. I don’t think the term “land-surface predictions” is widely understandable to the ICLR community, and it’s hard to see where the line is between these tasks and other tasks studied by the remote sensing foundation model community.
- I think the main novelty is in the selection of the downstream tasks and data sources. I appreciate that the performance is better on the downstream tasks, but this seems somewhat obvious because the input variables are similar to those being predicted in the downstream tasks. Other aspects of the approach have already been proposed in prior work. For example, the Cross-Variable Group Masking and point-based inputs (single pixel time series + static variables) was done in [Presto](https://arxiv.org/abs/2304.14065). Presto is also even smaller than StefaLand.
- Incomplete or unclear experiments
    - The paper uses PrithviWxC and TerraMind as baselines. It seems like any remote sensing foundation model could be used here. The two chosen are among the largest and lowest performance in the recent literature, so they seem like odd choices. What about models like [Galileo](https://arxiv.org/pdf/2502.09356), which also includes time-varying and static variables including some that are also in StefaLand (precipitation, soil moisture, evapotranspiration, etc), or Presto which is also attribute-based, or other recent models that have been shown to have higher performance than Prithvi (e.g. see comparisons in the Galileo paper).
    - The paper says they could not run PrithviWxC and TerraMind for all tasks due to computational expense, but the choice of when they are included seems arbitrary. I think the models were not available at the time of writing this paper, but now the authors could also try precomputed embeddings from [Tessera](https://arxiv.org/abs/2506.20380) or [AlphaEarth Foundations](https://arxiv.org/abs/2507.22291v1).
    - There is not an explicit ablation experiment in the paper. There seem to be versions of the model that would constitute ablations in the various tables (like StefaLand - no resConn) but this makes it confusing which are ablations and which are the actual proposed model. Which one is “StefaLand” versus an ablation? This is even more confusing because not all variants are included in all tables, for example Table 2 only has “StefaLand - direct”.
    - The tables mostly report means over cross-validation splits, but there are no standard errors, so we cannot assess the variability/stability of each model and the significance of the comparisons.
- The sources in Table C2 seem somewhat random. What was the motivation for this specific list, or excluding other sources that could have been used?
- The appendix referencing is coarse. For example, an entire section of the Appendix is reference where you need to reference a table in that section, so the reader has to look through the section to find the material referenced. This is a nit, but it makes it tedious to read.

**Questions:**

- are the downstream tasks predicting *future* states based on previous timesteps, like future soil characteristics or streamflow?
- I don’t think the term “land-surface predictions” is widely understandable to the ICLR community, and it’s hard to see where the line is between these tasks and other tasks studied by the remote sensing foundation model community.  Can you define “land-surface predictions” and explain the distinction from other tasks types in remote sensing?
- What was the reasoning for choosing PrithviWxC and TerraMind, as opposed to other choices that would be smaller, more performance, and/or more related in terms of the input sources to StefaLand?
- Can the authors add standard errors to the results tables?
- What was the motivation for this specific list of dataset sources, or excluding other sources that could have been used?

---

> ### Author Response · Authors · 2025-11-22
>
> Weakness responses
>
> 1. Thanks for the comments. The streamflow and soil moisture tasks here are dynamical sequence-to-sequence future prediction tasks because hydrological and land-surface systems are input driven systems. We need rainfall input today to predict today’s streamflow or soil moisture. This is the standard prediction setup. The model learns a response to the forcings plus some memory and modulation effects. For many scenarios we use reanalysis forcing, while we can also use forecasted weather for pure forecast. We rely on forecasted inputs, not autoregressive prediction of the land-surface variables themselves because this system will dry up without inputs.
>
> 2. To clarify,  by “land surface predictions”, we  refer to evolution of physical land-surface states and fluxes such as runoff generation, soil moisture dynamics, infiltration and evapotranspiration interactions, and surface energy fluxes. These variables evolve through coupled terrestrial processes and are represented as multi-feature time series rather than imagery or atmospheric fields.
>
>    Existing Earth foundation model families focus on different domains. Atmospheric and weather models such as GraphCast, Pangu Weather, Prithvi WxC, and FuXi Weather concentrate on atmospheric dynamics and predict pressure, wind, temperature, or related meteorological fields. Remote sensing foundation models such as Prithvi EO, SatMAE, Scale MAE, GeoCLIP, and RingMo focus on satellite imagery tasks including segmentation, land cover classification, change detection, and image reconstruction. Climate oriented ML foundation models target coarse climate variables or long term climate trajectories rather than terrestrial hydrology.
>
>    Across these atmospheric, climate, and remote sensing model families we were unable to identify foundation models that address land surface hydrologic processes such as runoff generation, infiltration-ET partitioning, soil moisture evolution, vegetation growth, or surface energy flux prediction. To our knowledge no existing Earth foundation models are designed for this class of land-surface dynamic problems.
>
> 3. You are right that some architectural elements have appeared in prior work.The novelty comes from showing that capturing landscape organization improves spatial generalization, that improving data relevance matters as much as increasing data volume, and that widely available scalar attributes contain the structural information needed for these terrestrial prediction tasks. These points shift the focus from model size and towards the structural information that matters for hydrology and soil systems. The downstream tasks were chosen to expose these relationships, not to engineer an easy benchmark, and they demonstrate that models benefit more from physically relevant inputs than from increasing complexity. The model is also intentionally small and researcher accessible.
>    From this perspective, the use of cross variable masking or point based encodings is not meant to claim architectural innovation but to build a setup that allows us to test those scientific ideas. Some components may be older in the ML literature, but they match the structure of input driven land surface systems and are appropriate for the problems we study.
>
>    One additional point is that the strength and consistency of this method's spatial generalization gains across four different tasks was a surprise, which shows the potential significance of these findings while still leaving room for future feature adaptations.
>
>
>
> 4. Unclear Experiments response
>
>    1. Thanks for the comments. This is not a remote sensing foundation model. The purpose of those comparisons was to demonstrate the gap between those models on this type of land surface prediction problem. TerraMind was listed according to the Pangea benchmark as a top rated model. We appreciate the suggestion with precomputed embeddings. Tessera embeddings are only available for 2024 and although they plan to release 2017 to 2024 these are not available yet. AlphaEarth has the same issue with embeddings only from 2017 to 2024\. This said, we did compare with the restricted time range for in situ soil moisture and soil composition where possible. See tables (above 5 and 6\) which show our model’s edge against AlphaEarth embeddings.
>
>
>
>    2. Sorry we didn’t use the word “ablation”, but several ablation tests were already there in the table and we also added linear probing of Stefaland in updated tables (above 2 and 4\) which are now clearly marked as ablations. Resconn is a time-series adaptation and was not used in soil composition or landslides as neither case made sense to use it. We will add Resconn to the global streamflow test.
>
>    3. This is a good suggestion. We added standard errors within cross validations that have been added to all experiments if applicable.

---

> > ### Author Response · Authors · 2025-11-22
> >
> > 5. Thanks for a chance to clarify. The motivation for selecting this list of datasets was to prioritize physically dominant, globally available, and consistently daily meteorological forcings, thereby ensuring fair comparability and reproducibility across tasks and regions. Variables  such as precipitation, temperature, radiation, and humidity were specifically included because they directly control the energy and water balance processing (e.g., infiltration, runoff, snowmelt, evapotranspiration) essential to multiple hydrological and landslide-related tasks. Additionally, remote sensing-derived attributes ( such as NDVI) were included as static or climatological characteristics to represent spatial variability effectively.
> >
> >    Conversely, other sources, particularly optical products, were excluded as dynamic inputs due to their irregular sampling intervals, frequent data gaps (caused by cloud cover, orbital revisit cycles), and shorter historical coverage periods (typically limited after the year 2000), making them less suitable for long-term daily alignment with target variables.
> >
> >
> > 6. Thank you for pointing this out. We will fix the appendix referencing and be more precise.
> >
> > Questions responses
> >
> > 1. You are right that none of the downstream tasks directly forecast future states without future inputs. As stated earlier, all of the hydrologic and land surface tasks are input driven systems.  The tasks here are considered standard future predictions (using reanalysis or forecast forcings). You need the future precipitation, temperature, and radiation to obtain the future streamflow or soil moisture. This is why it does not make physical sense to forecast these variables in the same way atmospheric or climate models do. The soil property task is an estimation problem. The land surface system predicts landslide susceptibility which is a likelihood of future events. It gives a measure of future risks from the present landscape. We will explain this in the revised manuscript if allowed.
> >
> > 2. There are major distinctions between the tasks demonstrated here, which include dynamic predictions and cross domain inferences, and remote sensing tasks, which seeks to retrieve information from data. The connection between our data and the predictand is indirect and only linked through dynamic process interactions and mutual shaping. The hydrologic tasks are about how the dynamic system responds and evolves. The landslide task estimates future baseline risks. The soil properties task is not temporal prediction but is inferring difficult-to-observe quantities. These missions require capturing linkages between processes, and they differ from remote sensing tasks that infer land cover or retrieve information from present data.
> >
> > 3. We sought models that have been discussed and advertised more.
> >    We appreciated the suggestion and we are currently working on adding comparisons with Galileo and Presto. Just as well, we have used AlphaEarth embeddings where applicable given its restricted date range and our model showed advantages. However, as the reviewer may understand, there is a nontrivial cost to run each of these models.
> >
> > 4. This has been done. See the tables provided.
> >
> > 5. Described above.
> >
> > We hope you see we sincerely worked hard to address your comments. We added new experiments, added many more benchmark models including AlphaEarth. We made clarification that this model focuses on dynamical predictions and cross-domain process interactions, and are fundamentally different in purpose and learned features from remote sensing foundation models. If you have more questions, please let us know.

---

> ### Author Response · Authors · 2025-11-28
>
> We accommodated reviewers' request for comparison with Galileo (Tseng et al., 2025). We attempted to compare the most natively applicable case which did require some custom adaptations to input layers. Since the expected input is multispectral time-series data and our landslide dataset consists of multichannel static attributes (17 channels, each 32×32 pixels). To accommodate this difference, we designed an adaptation strategy: first, we apply average pooling along the height dimension of the spatial features, then map the 17 environmental feature channels to 12 channels via a linear projection layer to match Galileo’s required input format. Concurrently, the width dimension is treated as pseudo-temporal steps. Specifically, we loaded Galileo’s pretrained Transformer encoder (with a 768-dimensional embedding space, positional encoding, and 4 attention layers, each containing 12 attention heads, and a feed-forward network dimension of 3072). Following the encoder output, we added a two-layer classification head (768→256→1), incorporating a ReLU activation function and dropout (rate = 0.3), for binary landslide prediction. The entire model was fine-tuned end-to-end using the AdamW optimizer (learning rate = 1e-4, weight decay = 0.01), with a batch size of 128, for 1000 epochs on the same train-test split. Gradient clipping (max norm = 1.0) was employed to ensure training stability.
>
> The complete table for the landslide case with paper comparison and the added Galileo results is included below.
>
> Table 5
> | Model  (10m)      | Accuracy ↑ | Precision ↑ | Recall ↑ | F1 ↑  | ROC AUC ↑ |
> |--------------------|------------|-------------|----------|-------|-----------|
> | Logistic Regression 2D | 0.744 | 0.720 | 0.795 | 0.756 | 0.823 |
> | Random Forest 2D       | 0.765 | 0.737 | 0.822 | 0.777 | 0.849 |
> | CNN2D                  | 0.880 | 0.896 | 0.858 | 0.877 | 0.854 |
> | Galileo + CNN2D        | 0.750 | 0.764 | 0.720 | 0.742 | 0.834 |
> | StefaLand + CNN2D      | 0.903 | 0.859 | 0.963 | 0.908 | 0.911 |
>
> Reference
> Tseng, G., Fuller, A., Reil, M., Herzog, H., Beukema, P., Bastani, F., Green, J. R., Shelhamer, E., Kerner, H., and Rolnick, D. (2025). Galileo: Learning Global and Local Features of Many Remote Sensing Modalities. arXiv:2502.09356.

---

### Author Response · Authors · 2025-11-22
**Revised Tables**

### Table 1 CAMELS streamflow performance
**PUB: random spatial holdout (ungauged basins)**

| Model              | RMSE | ubRMSE | Corr | NSE |
|--------------------|------|--------|------|-----|
| LSTM SL            | 1.402 ± 0.0389 | 1.360 ± 0.0388 | 0.762 ± 0.0064 | 0.636 ± 0.0386 |
| DLinear            | 2.012 ± 0.0628 | 2.000 ± 0.0646 | 0.598 ± 0.0083 | 0.302 ± 0.4840 |
| Informer           | 2.262 ± 0.0689 | 2.237 ± 0.0754 | 0.521 ± 0.0122 | 0.104 ± 0.0813 |
| Reformer           | 1.908 ± 0.0854 | 1.871 ± 0.0890 | 0.718 ± 0.0049 | 0.270 ± 0.1843 |
| TerraMind resConn  | 1.332 ± 0.0410 | 1.301 ± 0.0375 | 0.777 ± 0.0071 | 0.637 ± 0.0431 |
| **StefaLand resConn** | **1.111 ± 0.0378** | **1.068 ± 0.0374** | **0.869 ± 0.0067** | **0.717 ± 0.1600** |
**PUR: regional holdout (ungauged regions)**

| Model              | RMSE | ubRMSE | Corr | NSE |
|--------------------|------|--------|------|-----|
| LSTM SL            | 1.609 ± 0.2399 | 1.457 ± 0.2170 | 0.743 ± 0.0195 | 0.554 ± 0.1345 |
| DLinear            | 2.019 ± 0.3496 | 1.983 ± 0.3381 | 0.597 ± 0.0321 | 0.290 ± 0.9889 |
| Informer           | 2.332 ± 0.4058 | 2.295 ± 0.3731 | 0.497 ± 0.0138 | 0.046 ± 0.2715 |
| Reformer           | 2.074 ± 0.3309 | 1.978 ± 0.3115 | 0.686 ± 0.0151 | 0.257 ± 1.1680 |
| TerraMind resConn  | 1.420 ± 0.2021 | 1.398 ± 0.1932 | 0.763 ± 0.0172 | 0.551 ± 0.1891 |
| **StefaLand resConn** | **1.344 ± 0.2097** | **1.334 ± 0.1873** | **0.801 ± 0.0220** | **0.635 ± 0.2460** |

### Table 2 StefaLand ablation study on CAMELS
**PUB: random spatial holdout**

| Variant               | RMSE | ubRMSE | Corr | NSE |
|-----------------------|------|--------|------|-----|
| StefaLand direct      | 1.882 ± 0.0901 | 1.849 ± 0.0691 | 0.538 ± 0.0105 | 0.395 ± 0.1431 |
| StefaLand scratch     | 1.355 ± 0.0394 | 1.332 ± 0.0368 | 0.801 ± 0.0031 | 0.661 ± 0.0372 |
| StefaLand noResConn   | 1.171 ± 0.0325 | 1.154 ± 0.0319 | 0.823 ± 0.0022 | 0.706 ± 0.3260 |
| StefaLand linear      | 1.452 ± 0.0542 | 1.366 ± 0.0533 | 0.751 ± 0.0127 | 0.661 ± 0.0430 |
| **StefaLand resConn** | **1.111 ± 0.0378** | **1.068 ± 0.0374** | **0.869 ± 0.0067** | **0.717 ± 0.1600** |
**PUR: regional holdout**

| Variant               | RMSE | ubRMSE | Corr | NSE |
|-----------------------|------|--------|------|-----|
| StefaLand direct      | 1.982 ± 0.4012 | 1.949 ± 0.3871 | 0.230 ± 0.0784 | 0.201 ± 1.2201 |
| StefaLand scratch     | 1.516 ± 0.3723 | 1.378 ± 0.3496 | 0.771 ± 0.0211 | 0.560 ± 0.3120 |
| StefaLand noResConn   | 1.376 ± 0.1987 | 1.356 ± 0.1712 | 0.798 ± 0.0195 | 0.610 ± 0.1345 |
| StefaLand linear      | 1.484 ± 0.2588 | 1.453 ± 0.2402 | 0.672 ± 0.0391 | 0.542 ± 0.3100 |
| **StefaLand resConn** | **1.344 ± 0.2097** | **1.334 ± 0.1873** | **0.801 ± 0.0220** | **0.635 ± 0.2460** |

### Table 3 Soil moisture prediction across ISMN
**Random location holdout**

| Model                           | RMSE | ubRMSE | Corr |
|----------------------------------|-------|--------|-------|
| LSTM SL                          | 0.073 ± 0.0018 | 0.055 ± 0.0008 | 0.764 ± 0.0062 |
| DLinear                          | 0.088 ± 0.0012 | 0.065 ± 0.0007 | 0.612 ± 0.0068 |
| Informer                         | 0.102 ± 0.0019 | 0.082 ± 0.0002 | 0.232 ± 0.0119 |
| Reformer                         | 0.090 ± 0.0022 | 0.071 ± 0.0008 | 0.568 ± 0.0146 |
| **StefaLand resConn (proposed)** | **0.068 ± 0.0013** | **0.054 ± 0.0004** | **0.783 ± 0.0054** |
| TerraMind resConn                | 0.083 ± 0.0021 | 0.062 ± 0.0007 | 0.694 ± 0.0289 |
| PrithviWxC resConn               | 0.081 ± 0.0019 | 0.060 ± 0.0004 | 0.703 ± 0.0390 |

**Regional holdout (Europe)**

| Model                           | RMSE | ubRMSE | Corr |
|----------------------------------|-------|--------|-------|
| LSTM SL                          | 0.112 | 0.053 | 0.510 |
| DLinear                          | 0.093 | 0.051 | 0.623 |
| Informer                         | 0.088 | 0.063 | 0.358 |
| Reformer                         | **0.087** | **0.058** | 0.553 |
| StefaLand resConn (proposed) | 0.090 | 0.059 | **0.638** |
| TerraMind resConn                | 0.101 | 0.080 | 0.519 |
| PrithviWxC resConn               | 0.103 | 0.079 | 0.523 |

---

> ### Author Response · Authors · 2025-11-22
>
> ### Table 4 StefaLand ablation study on soil moisture
> **Random location holdout**
>
> | Variant             | RMSE | ubRMSE | Corr |
> |---------------------|-------|--------|-------|
> | StefaLand direct    | 0.140 ± 0.0431 | 0.103 ± 0.0041 | 0.637 ± 0.0352 |
> | StefaLand scratch   | 0.074 ± 0.0011 | 0.058 ± 0.0003 | 0.749 ± 0.0172 |
> | StefaLand noResConn | 0.075 ± 0.0009 | 0.057 ± 0.0001 | 0.741 ± 0.0201 |
> | StefaLand Linear    | 0.084 ± 0.0010 | 0.061 ± 0.0002 | 0.720 ± 0.0192 |
> | **StefaLand resConn (proposed)** | **0.068 ± 0.0013** | **0.054 ± 0.0004** | **0.783 ± 0.0054** |
> **Regional holdout (Europe)**
>
> | Variant             | RMSE | ubRMSE | Corr |
> |---------------------|-------|--------|-------|
> | StefaLand direct    | 0.135 | 0.112 | 0.503 |
> | StefaLand scratch   | 0.108 | 0.064 | 0.528 |
> | StefaLand noResConn | 0.095 | **0.058** | 0.545 |
> | StefaLand Linear    | 0.100 | 0.063 | 0.393 |
> | StefaLand resConn (proposed) | **0.090** | 0.059 | **0.638** |
>
> ### Table  5 Soil composition prediction
> **Clay**
>
> | Target | Model               | Corr | R^2 |
> |--------|----------------------|-------|-------|
> | Clay | Random Forest         | 0.456 ± 0.0156 | 0.207 ± 0.0128 |
> | Clay | Linear Regression     | 0.138 ± 0.0097 | 0.019 ± 0.0052 |
> | Clay | StefaLand direct      | 0.197 ± 0.0084 | 0.038 ± 0.0101 |
> | Clay | **StefaLand finetune (proposed)** | **0.509 ± 0.0072** | **0.259 ± 0.0089** |
> | Clay | AlphaEarth Linear     | 0.387 ± 0.0310 | 0.146 ± 0.0280 |
> | Clay | AlphaEarth finetune   | 0.462 ± 0.0277 | 0.205 ± 0.0219 |
>
> **Sand**
>
> | Target | Model               | Corr | R^2 |
> |--------|----------------------|-------|-------|
> | Sand | Random Forest         | 0.585 ± 0.0064 | 0.342 ± 0.0072 |
> | Sand | Linear Regression     | 0.347 ± 0.0107 | 0.120 ± 0.0072 |
> | Sand | StefaLand direct      | 0.253 ± 0.0121 | 0.064 ± 0.0081 |
> | Sand | **StefaLand finetune (proposed)** | **0.704 ± 0.0091** | **0.495 ± 0.0107** |
> | Sand | AlphaEarth Linear     | 0.537 ± 0.0140 | 0.287 ± 0.0160 |
> | Sand | AlphaEarth finetune   | 0.615 ± 0.0090 | 0.373 ± 0.0100 |
>
> ### Table 6 Restricted soil moisture dataset 2017 to 2020
> **Random location holdout**
>
> | Model                     | RMSE | ubRMSE | Corr |
> |---------------------------|-------|--------|-------|
> | AlphaEarth no ResConn     | 0.083 ± 0.0011 | 0.066 ± 0.0010 | 0.265 ± 0.0333 |
> | AlphaEarth ResConn        | 0.076 ± 0.0007 | **0.060 ± 0.0003** | 0.447 ± 0.0175 |
> | StefaLand resConn (proposed) | **0.071 ± 0.0054** | 0.068 ± 0.0030| **0.504 ± 0.0800** |
> **Regional location holdout**
>
> | Model                     | RMSE | ubRMSE | Corr |
> |---------------------------|-------|--------|-------|
> | AlphaEarth no ResConn     | 0.083 | 0.067 | 0.116 |
> | AlphaEarth ResConn        | **0.072** | **0.055** | 0.207 |
> | StefaLand resConn (proposed) | 0.086 | 0.064 | **0.252** |

---

### Author Response · Authors · 2025-12-03
**Final Notes**

We appreciate the reviewers’ efforts and the helpful suggestions they provided. Because the OpenReview bug shortened the discussion period, only a portion of the expected exchanges could take place.

Two reviewers (UBGy and 6uDU) engaged during the limited discussion window.
 UBGy explicitly stated they would raise their score after seeing additional literature comparisons; we delivered those comparisons in detail.
 6uDU clarified that their remaining concerns centered on the need for either (i) additional mathematical justification of our architectural adaptations or (ii) broader comparison to potential foundation-model frameworks. In response, we provided a clear explanation of how the residual connector and the static–dynamic separation are tied to the structure of land-surface systems, and we highlighted the expanded set of comparisons we had already added. UBGy agreed that the work fits within “AI for Science,” given the novelty and relevance to an important application domain in a reply to 6uDU.

The reviewer with the lowest score, jBPF, did not have time to follow up. Despite their self-reported confidence level being the highest possible, several of their questions indicated a misunderstanding of core concepts relevant to this work, particularly the distinction between remote-sensing information retrieval and land-surface prediction, which depends on multidomain process interactions and dynamic system responses in hydrologic and landslide settings. We clarified that land-surface prediction has a long-established meaning in geoscience and that this work differs fundamentally from remote-sensing foundation models. We also explained that our tasks are input-driven physical prediction problems rather than autoregressive forecasting of the state variables themselves. We addressed every point in their initial review and produced additional comparisons they requested. The new experiments, including Galileo, AlphaEarth, the added standard errors, and the expanded ablations, consistently showed that our approach performs better. Since these results directly address the concerns raised, we believe their assessment would likely have changed had they been able to review this evidence.


We also want to restate a point that we already addressed in our initial reply to 6uDU regarding supposedly unreported baselines. Their comment referred to values shown in table 2 of the comparison paper at 10 meter resolution. In our submission, we noted that our reported numbers used the 30 meter resolution version of the dataset, which is one of the settings used in the published comparison. Nothing was omitted. After the issue was raised, we generated the corresponding 10 meter results for our method, and these results show even stronger performance. Thus the absence of the 10 meter comparison in the initial submission reduced the apparent strength of the model and was not an attempt to misrepresent its performance.

Finally, we note that there is currently no established earth system foundation model designed for land-surface applications. As our comparisons show, existing atmospheric or remote-sensing models do not transfer well to these tasks, while StefaLand provides consistent gains and is, to our knowledge, the only model that effectively serves this domain.

For the full revision, we will incorporate all reviewer-requested changes, including:

Additional experiments and comparisons (see tables posted below)

* Added experiment-level standard errors across all applicable tables

* Added linear probing ablations

* Split ablation results into clearer tables

* Added comparisons with AlphaEarth and Galileo

* Added DLinear, Informer, and Reformer baselines for the temporal tasks

* Added the 10 m landslide results to accompany the 30 m results

Clarity and presentation improvements

* Moved mathematical details of Stefaland to the appendix and replaced them with a visual diagram of StefaLand. Along with minor corrections to math suggested by UBGy

* Added the literature comparison tables requested by UBGy to appendix

* Clarified that our focus is on dynamically forced terrestrial systems where spatial generalization is the primary challenge, and that the MAE structure is chosen to reflect land-surface prediction rather than image-based remote-sensing tasks

We put considerable effort into addressing all concerns despite the shortened discussion period, and the follow-up exchanges show that at least one reviewer explicitly intended to raise their score once additional comparisons were provided and another reviewer clarified that their remaining concerns were specific and addressable. We hope the AC considers these updates and the incomplete discussion context when making a final decision.

---

### Meta-Review · Area_Chair_6mgz · 2026-01-10

**Summary:**

StefaLand contributes a model that is smaller and more efficient, for tasks that focus on the land surface and its change rather than static recognition (of land use, cover, ...), with results that improve on some larger foundation models of satellite imagery and other remote sensing data. The model is based on the popular masked auto-encoder (MAE) method and much of the efficiency is due to the choice of lower-dimensional attribute inputs rather than the higher-dimensional remote sensing inputs used for other foundation models.

Three expert reviewers recommend rejection. The key points are that the contributed model and experiments are not sufficiently clearly presented for a machine learning audience, nor are they adequately compared and related to existing work.

The substantial rebuttal replied to each review, commented with a general summary, and updated a number of results across the tables of the submission. The rebuttal also included a revision of the paper incorporating clarifications, additional measurements like standard errors, and add more precise references to material in the main text and appendix as suggested by reviewers. While the rebuttal is an improvement, it does not fully resolve the concerns, and the necessary changes are enough to merit another round of review because they are many and involve key details of the evaluation and comparison.

**Reviewer Concerns:**

- Insufficient evaluation and baselines (jPBF, UBGy, 6uDU): The chosen Prithvi and TerraMind baselines are some of the largest and lowest performing models, while others like Galileo and Presto are more related to StefaLand in their use of point-wise inputs. There are no published baselines for all but one tasks, and while an effort is made to evaluate other models, it is not clear how well they are applied or if other choices (like Galileo as mentioned) would do better. The pre-training of StefaLand is only evaluated by fine-tuning, instead of by linear probing and nearest neighbors too, which is not typical for proposed "foundation" models. In the rebuttal the authors add results for Galileo, but with potentially unsound design choices that do not reflect that Galileo can also accept static inputs), and for more transformers (Informer and Reformer), and they do not improve on StefaLand or the contributed LSTM baseline. Partially resolved, due to more baselines, but not fully resolved, due to more work being needed to explore and explain the use of existing models. This work requires another round of review, because the details matter for evaluation.
- Insufficient clarity and connection to machine learning and remote sensing as a topic and audience (jPBF, UBGy, 6uDU): The presentation is not clear enough in terminology and tasks for a machine learning audience such as ICLR. Which configuration of the proposed model is _the_ method and which are ablations? What are the guidelines for when to use which? Should this be proposed as a pre-trained model (for fine-tuning), in which case experiments should compare to StefaLand from scratch vs. StefaLand pre-trained on each downstream task, or as a foundation model with more types of transfer (fine-tuning, linear probing, retrieval). Unresolved.
- Insufficient novelty and methodological contribution (6uDU): The proposed StefaLand can be interpreted as a combination of feature engineering and the MAE model applied to geological time-series data. As such, this makes it an application of existing modeling rather than a contribution to this type of modeling itself, especially given the limited scope of types of self-supervision in only trying reconstruction by MAE and given the exclusion of other methods for time-series modeling. The reviewer and authors discuss, and the meta-reviewer considers this point resolved: the submission is in scope for ICLR as an applied AI for science project. Nevertheless it must still satisfy standards for experimentation, related work, and clarity like all other submissions.
- Confusion about the tasks (jPBF, 6uDU): Is this forecasting, change detection/difference prediction, or sequence-to-sequence prediction? This is not adequately explained by the text, but it is explained by the rebuttal, which confirms the proposed tasks and model are not forecasting. The confusion is resolved, in that the tasks are not forecasting, but the exclusion of other models (like Galileo and Presto which are likewise not for forecasting) is therefore unresolved.
- Comparison with precomputed embeddings (jPBF): The authors provide a few results for downstream tasks with AlphaEarth embeddings. However, these are not entirely clear because there is a table row for "fine-tuning" which is not possible for this method given that it is released as embeddings data and not a model for further optimization. Nevertheless the meta-reviewer acknowledges and appreciates the effort to compare with a method suggested by the reviewer. (Note: these were not available a the time of submission, as acknowledged by the reviewer, so the meta-reviewer wishes to make clear that any response here is only a positive and cannot be counted as a negative.) Resolved.

**Reviewer Scores:**

- jPBF: The reviewer could maintain their score of 2. The main concerns of this reviewer relate to the fuller and comparable benchmarking of the proposed StefaLand with existing Earth foundation models and data sources, and while an attempt was made in this direction, these are not resolved. While Galileo and AlphaEarth were tried, the Galileo application is unusual w.r.t. to the use of Galileo in its own experiments (in which both static and dynamic inputs are used for downstream tasks) and unclear in its use of AlphaEarth.
- UBGy: The reviewer would most likely raise their score to at least 5. Key points on the insufficiency of baselines and potential incomparability of evaluations were resolved by the rebuttal providing providing more results and reporting results side-by-side with those published in previous papers.
- 6uDU: The reviewer could maintain their 4 or raise to 5. Discussion indicates the reviewer initially remains unconvinced by the arguments for the work as an AI4Science paper, but the further experiments and detailed clarifications may be partially persuasive. All the same these multiple posts of clarifications would need to be incorporated into the main text.

---

### Decision · Program_Chairs · 2026-01-26

Reject